# Repaired Tetralogy of Fallot Pressure Assessment: Insights from 4D-Flow Pressure Mapping

**Safia Ihsan Ali** [1,2], **David Patton** [3,4], **Kimberley A. Myers** [3] and **Julio Garcia** [2,4,5,6,*]

1. Department of Biomedical Engineering, University of Calgary, Calgary, AB T2N 1N4, Canada; safia.ihsanali@ucalgary.ca
2. Stephenson Cardiac Imaging Centre, University of Calgary, Calgary, AB T2N 1N4, Canada
3. Department of Pediatrics, Cumming School of Medicine, University of Calgary, Calgary, AB T2N 1N4, Canada; david.patton@albertahealthservices.ca (D.P.); kim.myers@albertahealthservices.ca (K.A.M.)
4. Department of Cardiac Sciences, Cumming School of Medicine, University of Calgary, Calgary, AB T2N 1N4, Canada
5. Libin Cardiovascular Institute, University of Calgary, Calgary, AB T2N 1N4, Canada
6. Department of Radiology, Cumming School of Medicine, University of Calgary, Calgary, AB T2N 1N4, Canada
* Correspondence: julio.garciaflores@ucalgary.ca

**Abstract:** Tetralogy of Fallot (TOF) is the most prevalent cyanotic congenital heart defect (CHD) that alters normal blood flow through the heart and accounts for 10% of all CHD. Pulmonary stenosis and regurgitation are common in adults who have undergone TOF repair (rTOF) and can impact the load on the right ventricle, blood flow pressure, and pulmonary hemodynamics. Pressure mapping, obtained through 4D-flow magnetic resonance imaging (4D-flow MRI), has been applied to identify abnormal heart hemodynamics in CHD. Hence, the aim of this research was to compare pressure drop and relative pressures between patients with repaired TOF (rTOF) and healthy volunteers. An in vitro validation was performed, followed by an in vivo validation. We hypothesized that pressure drop is a more stable pressure mapping method than relative pressures to detect altered hemodynamics. A total of 36 subjects, 18 rTOF patients and 18 controls underwent cardiac MRI scans and 4D-flow MRI. Pressure drops and relative pressures in the MPA were higher in rTOF patients compared to the controls ($p < 0.05$). Following the in vitro validation, pressure drops proved to be a more stable pressure mapping method than relative pressures, as the flow loses its laminarity and becomes more turbulent. In conclusion, this study demonstrated that flow hemodynamics in rTOF can exhibit altered pressure maps. Pressure mapping can help provide further insight into rTOF patients' hemodynamics to improve patient care and clinical decisions.

**Keywords:** repaired tetralogy of Fallot; magnetic resonance imaging; 4D-flow MRI; heart hemodynamics; pressure mapping





## 1. Introduction

The cardiovascular system keeps blood flowing efficiently to achieve laminar flow through the vessels and the chambers of the heart. In numerous cases of congenital heart disease, particularly in individuals with surgically repaired tetralogy of Fallot (rTOF), there is evidence of nonlaminar flow in the right ventricle (RV), which includes transitional and turbulent flow [1,2]. TOF is the most prevalent cyanotic congenital heart defect (CHD) that alters normal blood flow through the heart and accounts for 10% of all CHD. TOF is characterized by a ventricular septal defect (VSD), aortic override, RVOT obstruction, and right ventricular hypertrophy. Surgical repair, which includes VSD closure and RVOT reconstruction, is typically performed in infancy [3,4]. The primary repair encompasses the closure of the VSD, removal of the obstructive infundibular muscle, and alleviation of the pulmonary stenosis. Patch reconstructions are typically used to repair a narrowed pulmonary artery [5]. Despite having improved clinical outcomes, patients with rTOF

require clinical and imaging follow-up to evaluate for post-surgical problems. The most common complications in patients with rTOF are pulmonary regurgitation (PR) and/or residual or reoccurring pulmonary stenosis (PS). PR, when severe, causes right ventricular dilatation, dysfunction, and arrhythmia. To prevent irreversible right ventricular failure, pulmonary valve replacement is then necessary [6].

Echocardiography and cardiac magnetic resonance (CMR) are the imaging modalities of choice for the follow-up of TOF patients. Both modalities can assess a wide range of anatomical and functional parameters, but both also have several limitations. Echocardiography can provide information on anatomy and physiology, while color Doppler can perform qualitative flow assessment. The quality of the image is greatly influenced by the acoustic window as well as the proficiency of the operator. CMR has become common in managing many congenital conditions, owing to its ability to visualize structures not well seen by echocardiography without ionizing radiation. It can provide a non-invasive three-dimensional cardiovascular anatomy, volumes, and function evaluation. In patients with rTOF, CMR has emerged as the imaging method of choice, playing an essential role in postoperative follow-up and evaluation [6]. Flow can be analyzed using two-dimensional phase-contrast (2D PC) MRI, providing flow volumes and velocity measurements perpendicular to a single plane placed in the vessel of interest. Each plane of interest must be individually planned to obtain flow measurements, and separate breath-holding scans must be performed. Furthermore, 2D PC MRI encounters challenges in accurately quantifying flow due to the heart's motion in relation to the imaging plane. It may provide an incomplete evaluation of blood flow due to technical limitations, particularly in cases of complex CHDs.

Recently, four-dimensional flow MRI (4D-flow MRI) has emerged as a promising and non-invasive imaging technique that can provide a comprehensive quantitative evaluation of flow in an entire volume within the chest in a single short imaging session [7]. That is, 4D-flow MRI illustrates 3D blood flow patterns and hemodynamics by utilizing velocity encoding (VENC) in three spatial directions. This imaging technique enhances our comprehension of blood flow properties in both normal and pathological conditions, offering comprehensive 3D visualization of anatomy and velocity. Consequently, it facilitates precise measurements of vessel lengths and provides valuable hemodynamic data [3]. This technique evaluates intricate flow patterns, such as helical or vortical flow, and measures advanced fluid dynamic parameters, including pressure difference maps, turbulent kinetic energy, and viscous energy loss [8]. Furthermore, 4D-flow MRI has been demonstrated to be effective for the qualitative and quantitative evaluation of pulmonary hemodynamics in TOF patients [9].

Blood pressure measurements play a crucial role in diagnosing cardiovascular disease and detecting irregular blood flow in large vessels [10]. Pressure gradients serve as significant clinical indicators for various cardiovascular conditions. In clinical practice, catheter measurements are considered the gold standard for assessing in vivo pressure gradients. Although this method is reliable and safe, it is an invasive procedure that carries potential complications and involves exposure to radiation for catheter guidance. An alternative approach for estimating pressure gradients is to utilize the simplified Bernoulli equation, which relies on the measurement of maximum velocity ($V_{max}$) obtained from standard clinical Doppler ultrasound (US). However, $V_{max}$ measurements using ultrasound are prone to error due to limited acoustic windows and technical difficulties in aligning Doppler interrogation with peak velocities. The exponentiation of velocities in the Bernoulli equation makes pressure difference estimation highly sensitive to errors in $V_{max}$ measurements. Another non-invasive and user-independent method for deriving relative pressure gradients is using time-resolved (CINE), three-directionally encoded phase contrast (PC) MRI to measure the time-resolved velocity field. By solving the Navier–Stokes (NS) equation, assuming blood is an incompressible, laminar Newtonian fluid, pressure gradients can be calculated accurately [11].

Furthermore, the NS equation can be transformed by taking the divergence into a pressure Poisson equation (PPE) [12]. Based on derived pressure gradients, pressure differences or pressure drops and relative pressure fields can be obtained by solving the PPE. For example, Bock et al. [11] applied the iterative approach to solving the PPE to obtain pressure differences or drops. On the other hand, Ebbers and Farnebäck [12] suggested using a multigrid-based PPE solver to compute relative pressures. However, no study to date has compared the iterative approach by Bock et al. [11] with the multigrid-based solver approach introduced by Ebbers and Farnebäck [12].

In addition to different pressure mapping methods, several ways exist to report these pressures. Numerous studies use 2D analysis planes positioned in the vessel of interest [13,14]. Some studies report the summation of pressures over the total volume [1,15]. In comparison, others report pressure values along a centerline that mimics a virtual catheter [16]. Many authors have used volumes and centerlines in literature, but no study has reported pressures using both. Therefore, this study aimed to compare pressure mapping methods and investigate whether one is more stable. We hypothesized that pressure drop is a more stable pressure mapping method than relative pressures to detect altered hemodynamics.

## 2. Materials and Methods

In vitro validation using a stenosis phantom was performed, followed by an in vivo validation between (i) healthy subjects and patients with rTOF who reported using volumes and centerlines and (ii) volume and centerline data of the controls and patients. The study diagram is presented in Figure 1.

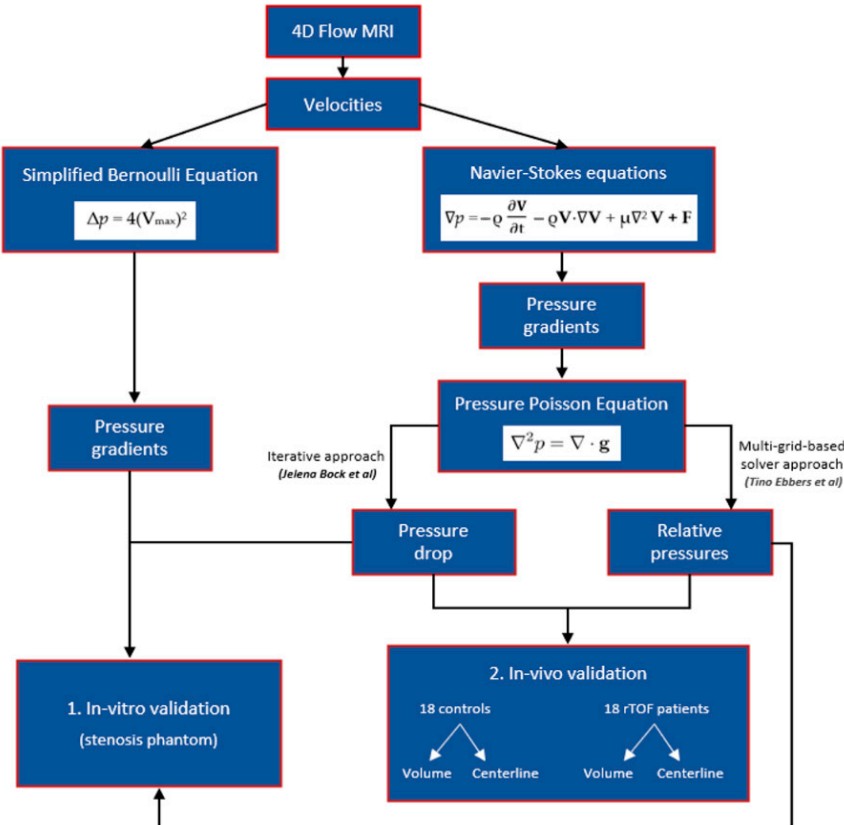

**Figure 1.** A schematic representation of the study. This figure illustrates the methodology employed in the study to estimate pressure gradients and derive pressure drops and relative pressure fields. Velocity measurements were used to estimate pressure gradients through the simplified Bernoulli and Navier–Stokes equations. The Navier–Stokes equation was further transformed into a pressure

Poisson equation (PPE) by taking the divergence. Pressure drops were obtained using an iterative approach to solving the PPE, as suggested by Bock et al. [11]. Ebbers and Farnebäck [12] proposed a multigrid-based PPE solver for computing relative pressures. The study involved in vitro validation followed by in vivo validation to compare these different pressure mapping methods. In vitro validation was initially performed using a stenosis phantom. Subsequently, in vivo validation was conducted, which involved comparing (i) healthy subjects and patients with repaired tetralogy of Fallot (rTOF) using volumes and centerlines and (ii) volume and centerline data of the controls and patients.

### 2.1. Pressure Gradients, Pressure Drop, and Relative Pressures Estimation

Pressure gradients $\nabla p = (\frac{\partial p}{\partial x}, \frac{\partial p}{\partial y}, \frac{\partial p}{\partial z})$ assuming a viscous, incompressible fluid was calculated from the three-directional velocity vector fields $\mathbf{V} = (u, v, w)$ using the NS equations:

$$\nabla p = -\rho \frac{\partial \mathbf{V}}{\partial t} - \rho \mathbf{V} \cdot \nabla \mathbf{V} + \mu \nabla^2 \mathbf{V} + \mathbf{F} \tag{1}$$

where $p$ is the pressure, $\rho$ is the fluid density, $\mathbf{V}$ is the measured three-directional velocity, $\mu$ is the dynamic fluid viscosity, and $\mathbf{F}$ represents the body forces. The terms on the right-hand side from left to right are transient inertia, convective inertia, viscous resistance, and body forces, respectively. As phantom and human subjects were placed in a horizontal position in the scanner, the body forces were neglected for all calculations.

By taking the divergence, the NS equation can be transformed into the PPE [12]:

$$\nabla^2 p = \nabla \cdot \mathbf{g} \tag{2}$$

where $\mathbf{g}$ denotes the pressure gradient field obtained from the right-hand side of Equation (1) while dropping the body force term. Pressure drop was calculated by applying the approach used by Bock et al. [11], where the pressure is initialized by region-growing integration of the pressure gradient with a user-defined reference point ($\Delta p = 0$), and all calculations were performed independently for each time frame. On the contrary, relative pressures were computed using the approach introduced by Ebbers and Farnebäck [12]. This method did not require a user-defined reference point and utilized a multigrid-based PPE solver that works directly on the structure-defined computational domain. For pressure gradient estimation, blood properties were set to $3.2 \times 10^{-3}$ Pa s and $1060$ kg/m$^3$ for viscosity and density, respectively. The pressure gradient $\Delta p$ across the stenosis in the in vitro model was determined by applying the simplified Bernoulli equation, as used in clinical routine [17]:

$$\Delta p = 4(V_{max})^2 \tag{3}$$

$V_{max}$ represents the peak velocity at the maximum narrowing of the stenosis of the phantom.

### 2.2. In Vitro Model

A simple stenosis phantom ($\varnothing_1$ $20.4 \pm 0.5$ mm, stenosis $\varnothing_2$ $10 \pm 0.5$ mm, EOA = $0.78$ cm$^2$) with a contraction coefficient = 1 (i.e., EOA/AVA, where the anatomic valve area (AVA) is the cross-sectional area of the stenosis phantom) by the potential flow theory was filled with glycerol and water, and connected to a pump under constant flow conditions (Chemflo Unit, MP Pumps, Inc., Fraser, MI, USA). To increase the signal-to-noise ratio (SNR), the fluid was doped with a contrast agent (Magnevist$^®$, Bayer Schering Pharma AG, Leverkusen, Germany) at a concentration of $1.08$ mmol/L. Then, 4D-flow measurements were performed on a 1.5 T System (Aera, Siemens AG, Erlangen, Germany). The measurement parameters were as follows: VENC = $1–4$ m/s along all three velocity encoding directions, spatial resolution $1.0 \times 1.0 \times 1.0$ mm$^3$, field of view (FOV) = $350 \times 350$ mm$^2$, flip angle = $15°$, TE/TR = $2.7–3.1/5.6–5.9$ ms, and scan time = 10 min.

*2.3. In Vivo Model*

2.3.1. Study Population

A total of 36 subjects, including 18 rTOF patients (age: $32 \pm 10$, 6 females) and 18 controls (age: $37 \pm 14$, 7 females), were recruited retrospectively. The reported values represent the group mean $\pm$ standard deviation. All subjects were registered with the Calgary Cardiovascular Imaging Registry (CIROC). The study was authorized by the board of the University of Calgary Research Ethics, and all subjects gave informed consent. The research activities were conducted in accordance with the Helsinki Declaration. Commercial software (cardioDITM, Cohesic Inc., Calgary, AB, Canada) was used to manage the study's routine capture of patient informed consent, health questionnaires, and standardized collection of MRI-related variables. Patients were required to meet the criteria of being $\geq 18$ years old and having a documented history of rTOF at the time of the examination to qualify for participation. However, patients with implantable devices, severe renal impairment (eGFR 30 mL/min/1.73 m$^2$), or other known contraindications for MRI were excluded [18]. In the control group, subjects were 18 years or older without any history of cardiovascular disease, diabetes, or uncontrolled hypertension. The controls who were unable to complete the MRI scan were also excluded. Before scanning, demographic data such as age, gender, height, weight, and heart rate were collected. The Mosteller formula converted volume and mass measurements to body surface area measurements.

2.3.2. Cardiac Magnetic Resonance Imaging Protocol

Cardiac imaging was examined utilizing 3T MRI scanners (Skyra and Prisma, Siemens, Erlangen, Germany) [18]. A standardized clinical imaging protocol was performed on all subjects, which involved retrospective electrocardiographic gating and time-resolved balanced steady-state free precession (SSFP) cine imaging of the LV in four-chamber, three-chamber, two-chamber, and short-axis views at end-expiration. In addition, a contrast-enhanced 3D magnetic resonance angiogram (MRA) of the cardiovascular structures was obtained using a gadolinium contrast volume of 0.2 mmol/kg (Gadovist$^®$, Bayer Inc., Mississauga, ON, Canada). For a period of 5–10 min following contrast administration, time-resolved three-dimensional phase-contrast MRI with three-directional velocity encoding and retrospective ECG-gating (known as 4D-flow, Siemens WIP 785A) was conducted to determine the in vivo blood flow velocities throughout the entire heart [1,15]. This whole-heart protocol has been previously described in our reports [1,19,20]. In brief, 4D-flow data was collected during free breathing using the navigator gating of diaphragmatic motion and the following sequence parameters were used: bandwidth = 455–495 Hz/Pixel; pulse repetition time = 4.53–5.07 ms; echo time = 2.01–2.35 ms; flip angle = 15 degrees, spatial resolution = 2.0–3.5 × 2.0–3.5 × 2.5–3.5 mm; temporal resolution = 25–35 ms; and VENC = 150–250 cm/s. The total acquisition time ranged from 5 to 10 min, depending on the heart rate and respiratory navigator efficiency. The number of phases varied from 25 to 30 depending on the clinical scan operator.

2.3.3. Cardiac Imaging and 4D-Flow Analysis

A blinded reader assessed standard cardiac images on the same day of acquisition using dedicated software, cvi42 version 5.11.5 (Circle Cardiovascular Imaging Inc., Calgary, AB, Canada). This was carried out to determine left and right end-diastolic volume (LVEDV; RVEDV), LV and RV end-systolic volume (LVSEV; RVESV), as well as LV and RV ejection fraction (LVEF; RVEF) [1,21].

Following the acquisition, all 4D-flow MRI data were pre-processed with the "Velomap tool", a MATLAB tool developed by Bock et al. in 2007 that is widely used in the flow MRI community [22]. The pre-processing tool was used to perform the following tasks: corrections for Maxwell terms, eddy currents, and aliasing (see Figure 3A). In addition, a 3D phase-contrast (PC) angiogram (PC MRA) was created after pre-processing (see Figure 3B). Using an in-house MATLAB-based tool called "4D-Flow Analysis Tool" [20], the angiogram was used to segment (see Figure 3C) the aorta (Ao), pulmonary artery (PA),

left ventricle (LV), and right ventricle (RV). A pressure mapping tool, "4D Flow Pressure Mapping Tool", created in MATLAB 2019a (MathWorks, Natick, MA, USA), was used to compute the pressure drop and relative pressures for each segmented vessel. Using the approach proposed by Elbaz et al. [16], 4D virtual catheter mathematical models for probing pressures were constructed as tubes with an automatically derived radius along the centerlines of the segmented vessels.

### 2.4. In Vitro Data Analysis

The in vitro data analysis was conducted using MATLAB 2019a (Mathworks, Natick, MA, USA) [23]. As shown in Figure 2, first, MRI velocities were used to calculate pressure gradients using the simplified Bernoulli equation. The pressure drop was then computed using the velocities by placing the reference point at the outflow tract. Then, the relative pressures were calculated from the velocities [11,12,24,25]. This procedure was followed for all VENCs (1–4 m/s). Finally, the influence of noise and filtering on both pressure mapping methods was investigated. The original pressure data were subjected to five levels of white Gaussian noise (SNR 5, 15, 30, 45, and 135) for each VENC, followed by a Gaussian filter. Additionally, the Reynolds number, Re = ρuD/μ, at the inlet and stenosis regions for each VENC (1–4 m/s), was calculated, where ρ is the density, u is the average velocity, D is the diameter, and μ is the dynamic viscosity [26].

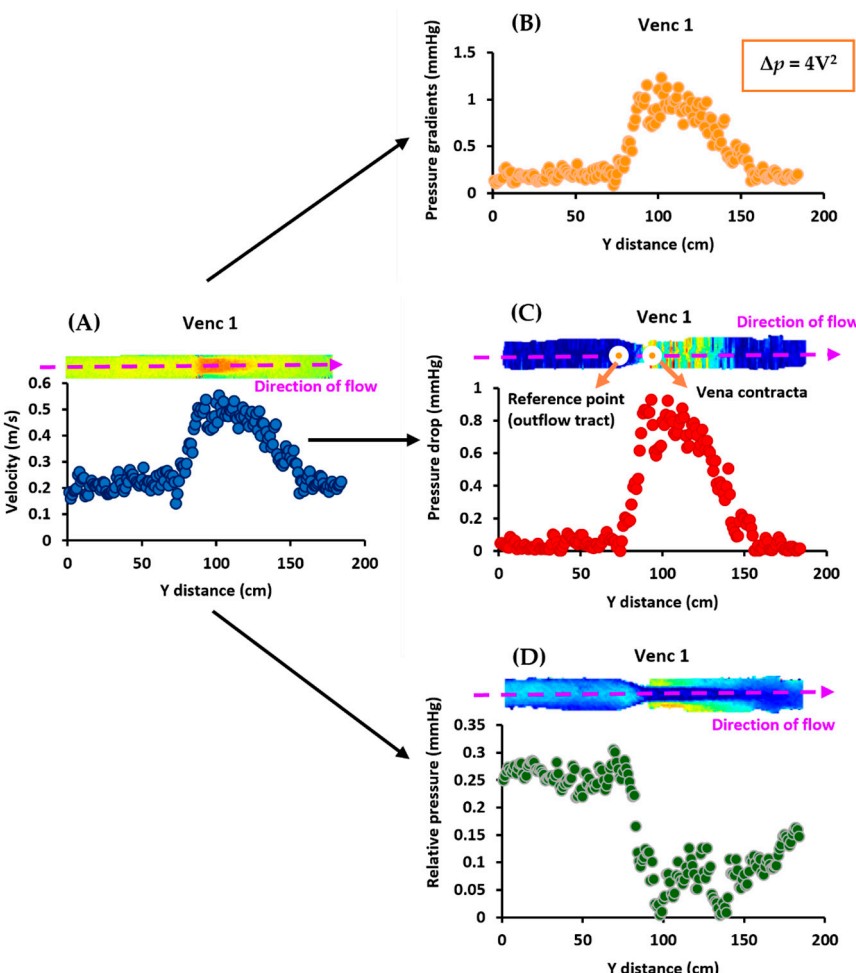

**Figure 2.** In vitro data analysis workflow. At VENC 1 (1 m/s), Panel (**A**) shows that the MRI velocities were first used to calculate (Panel (**B**)) the pressure gradients using the simplified Bernoulli equation, $\Delta p = 4V^2$. Next, they were used to compute the (Panel (**C**)) pressure drop by placing the reference point at the outflow tract. Finally, (Panel (**D**)) the relative pressures were calculated using the velocities. This process was repeated for the other VENCs (2–4 m/s).

## 2.5. In Vivo Data Analysis

The analysis was performed in MATLAB 2019a (Mathworks, Natick, MA, USA). As illustrated in Figure 3D, the reference points for the Ao, PA, LV, and RV were set at the left ventricular outflow tract (LVOT), RVOT, mitral valve, and tricuspid valve, respectively, for the pressure drop calculation. The pressure mapping tool divided the Ao into three sections: ascending aorta (AAo), aortic arch, and descending aorta (DAo). MPA, RPA, and LPA were the three divided sections of the PA. The entirety of the RV and LV was used for the evaluation. A variety of hemodynamic parameters were computed for each section at peak systole, including mean pressure drop (PDmean), mean relative pressure (RPmean), maximum pressure drop (PDmax), maximum relative pressure (RPmax), and standard deviation pressure drop (PDstd) and standard deviation relative pressure (RPstd). PDmean and RPmean were defined as the mean pressure drop and mean relative pressures in each region, respectively. Likewise, PDmax and RPmax were defined as the maximum pressure drop and maximum relative pressures in any voxel in each region, respectively. Finally, PDstd and RPstd were defined as the standard deviation pressure drop and standard deviation relative pressures in each region, respectively.

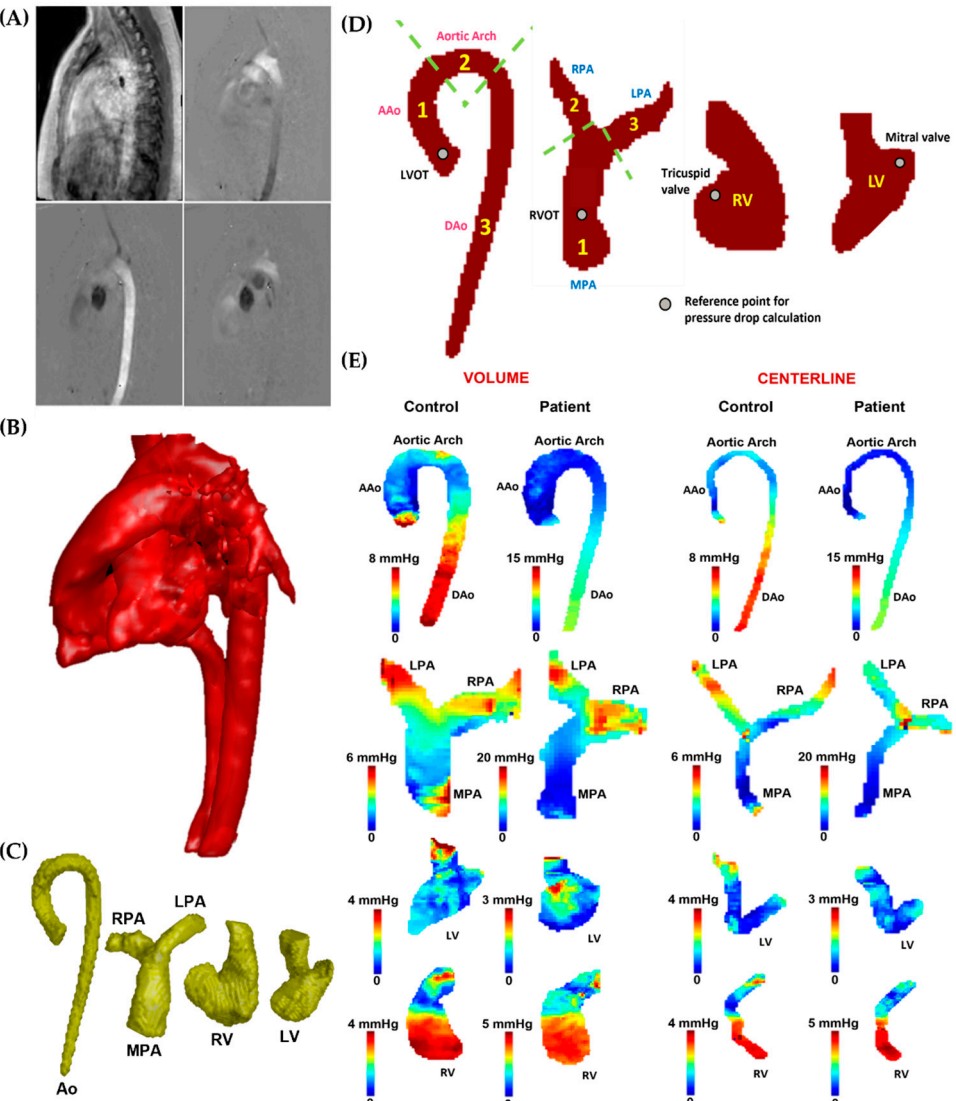

**Figure 3.** In vivo data analysis workflow. Panel (**A**): Initially, the 4D-flow velocity data in each direction (Vx, Vy, and Vz) were subjected to correction for eddy currents, noise, and aliasing. Subsequently, a phase contrast-magnetic resonance angiogram (PC-MRA) was generated (Panel (**B**)) to isolate the

specific vessels (Panel (**C**)), including the aorta (Ao), pulmonary artery (PA), left ventricle (LV), and right ventricle (RV). Panel (**D**): After segmentation, the aorta was divided into ascending aorta (AAo), aortic arch, and descending aorta (DAo), while the left ventricular outflow tract (LVOT) was used as the reference point for the pressure drop calculation. The PA was divided into the main pulmonary artery (MPA), right pulmonary artery (RPA), and left pulmonary artery (LPA); the right ventricular outflow tract (RVOT) was used as the reference point for the pressure drop calculation. The entirety of the LV and RV was used for the analysis; the mitral and tricuspid valves were used as reference points for the pressure drop calculation. Panel (**E**) illustrates the pressure drop measurements in the AAo, aortic arch, DAo, MPA, LPA, RPA, LV, and RV of the control and a patient, reported within vessel volumes and centerlines.

*2.6. Statistical Analysis*

The statistical analysis was conducted using SPSS 25 (SPSS, Chicago, IL, USA). To assess normality, the Shapiro–Wilk and Kolmogorov–Smirnov tests were utilized. However, since the data did not follow a normal distribution, a Mann–Whitney U test was performed for both study demographics, as well as for the pressure drop and relative pressure comparison between the two groups. The results are presented as the group mean $\pm$ standard deviation with a *p*-value < 0.05 to determine statistical significance. Furthermore, Spearman's correlation was used to analyze the relationship of pressure drop and relative pressures with the left ventricular ejection fraction (LVEF), right ventricular ejection fraction (RVEF), indexed left ventricular end-diastolic volume (LVEDVi), indexed left ventricular systolic volume (LVESVi), indexed right ventricular end-diastolic volume (RVEDVi), and indexed right ventricular end-systolic volume (RVESVi). The body surface area was utilized for indexation. A *p*-value < 0.01 was considered statistically significant. Furthermore, scatter plots were employed for each VENC to investigate the impact of the five levels of Gaussian noise and filters on both pressure mapping methods. Finally, Bland–Altman plots were used for each VENC to assess the differences between original pressure data and data with all five Gaussian noise levels, as well as between original pressure data and data with the Gaussian filter.

## 3. Results

### 3.1. In Vitro Data

Pressure gradients calculated using the simplified Bernoulli equation followed the same profiles as MRI velocities, as shown in Figure 4. Furthermore, the pressure drop profiles followed the trends of the velocities and pressure gradients, with the maximum pressure drop occurring at the location of the vena contracta. Pressure drop and pressure gradients produced similar results for each VENC. Relative pressures, on the other hand, did not follow the profiles of velocities, pressure drop, or pressure gradients; instead, different results were observed for each VENC. Supplementary Materials presents the scatter and Bland–Altman plots, illustrating the impact of Gaussian noise and filters on pressure drop and relative pressures. Finally, Re for each VENC (1–4 m/s) at the inlet and stenosis regions were 1148 and 1225, 2500 and 2683, 3243 and 3776, and 3784 and 4770, respectively.

### 3.2. Patient Characteristics

Table 1 shows the clinical parameters as well as the demographic data collected for the 18 patients and 18 controls who participated in the study. The controls had a higher age at the scan than patients (37 $\pm$ 14 years vs. 32 $\pm$ 10 years, *p* = 0.39). As displayed in Table 1, RVEF (56 $\pm$ 4% vs. 46 $\pm$ 10%, *p* = 0.00), RVEDV (172 $\pm$ 60 mL vs. 253 $\pm$ 95 mL, *p* = 0.02), RVEDVi (88 $\pm$ 20 mL/m$^2$ vs. 135 $\pm$ 47 mL/m$^2$, *p* = 0.00), RVESV (77 $\pm$ 31 mL vs. 142 $\pm$ 74 mL, *p* = 0.01), and RVESVi (39 $\pm$ 12 mL/m$^2$ vs. 76 $\pm$ 39 mL/m$^2$, *p* $\leq$ 0.001) were statistically significantly different between the patients and controls. As expected, rTOF patients had significantly larger RV volumes and lower RVEF compared with healthy controls.

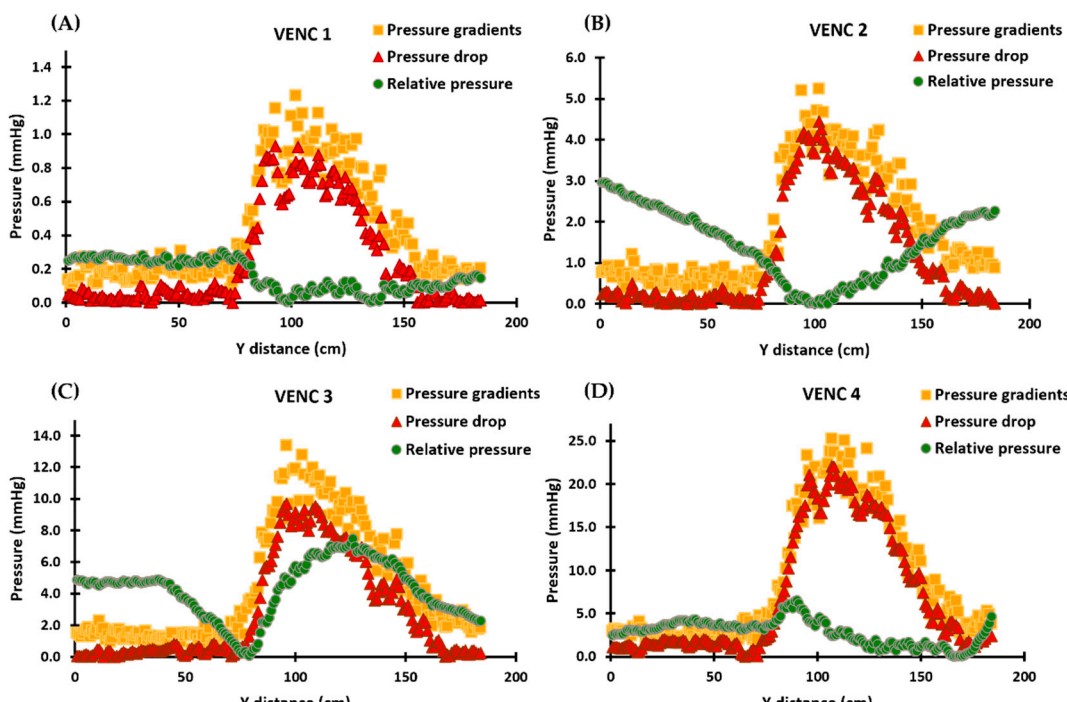

**Figure 4.** In vitro validation results demonstrating the calculated pressure gradients, pressure drop, and relative pressures for each VENC. Panels (**A**–**D**): The profiles of the pressure gradients are calculated using the simplified Bernoulli equation and are similar to those of MRI velocities. The pressure drop profiles follow the velocities and pressure gradients, with the maximum pressure drop occurring at the vena contracta location. The relative pressures, however, do not follow the velocity, pressure drop, or pressure gradient profiles and instead align with the pressure recovery phenomenon in Panels (**A**,**B**). In Panels (**C**,**D**), the relative pressures no longer align with the pressure recovery phenomenon and exhibit increased instability. For each VENC, the pressure drop and pressure gradient results show comparable results, while the profiles of relative pressures undergo a significant change and become increasingly unstable.

**Table 1.** Subject data baseline characteristics.

| Characteristic | Patients (*n* = 18) | Controls (*n* = 18) | *p*-Value |
|---|---|---|---|
| Age at scan (year) | $32 \pm 10$ | $37 \pm 14$ | 0.39 |
| Sex (f/m) | 6/12 | 7/11 | 0.73 |
| BSA (m$^2$) | $1.87 \pm 0.19$ | $1.88 \pm 0.32$ | 0.83 |
| HR (bpm) | $73 \pm 10$ | $67 \pm 12$ | 0.08 |
| BP systolic (mmHg) | $109 \pm 8$ | $107 \pm 12$ | 0.63 |
| BP diastolic (mmHg) | $60 \pm 10$ | $58 \pm 12$ | 0.40 |
| LVEF (%) | $58 \pm 9$ | $61 \pm 3$ | 0.11 |
| LVEDV (mL) | $151 \pm 41$ | $132 \pm 49$ | 0.14 |
| LVEDVi (mL/m$^2$) | $80 \pm 17$ | $110 \pm 48$ | 0.09 |
| LVESV (mL) | $65 \pm 26$ | $52 \pm 20$ | 0.10 |
| LVESVi (mL/m$^2$) | $35 \pm 13$ | $43 \pm 19$ | 0.26 |
| RVEF (%) | $46 \pm 10$ | $56 \pm 4$ | 0.00 |
| RVEDV (mL) | $253 \pm 95$ | $172 \pm 60$ | 0.02 |
| RVEDVi (mL/m$^2$) | $135 \pm 47$ | $88 \pm 20$ | 0.00 |
| RVESV (mL) | $142 \pm 74$ | $77 \pm 31$ | 0.01 |
| RVESVi (mL/m$^2$) | $76 \pm 39$ | $39 \pm 12$ | <0.001 |

BSA: body surface area; HR: heart rate; BP: blood pressure; LVEDVi: indexed left ventricular end diastolic volume; LVESVi: indexed left ventricular end systolic volume; LVEF: left ventricular ejection fraction; LVEDV: left ventricular end diastolic volume; LVESV: left ventricular end diastolic volume; RVEDVi: indexed right ventricular end diastolic volume; RVESVi: indexed right ventricular end systolic volume; RVEF: right ventricular ejection fraction; RVEDV: right ventricular end diastolic volume; RVESV: right ventricular end systolic volume.

*3.3. In Vivo Data*

Mean, max, and STD pressure measurements of the aorta, LV, PA, and the RV reported as volumes are shown in Figure 5. For the left side of the heart, the controls demonstrated a higher PDmean in the AAo, aortic arch, DAo, and LV. Statistically significant differences between the two cohorts for PDmean were only found in the AAo ($3.017 \pm 1.461$ mmHg vs. $1.928 \pm 0.990$ mmHg, $p < 0.05$). On the other hand, a higher RPmean was seen in the AAo and the LV of the controls. However, a higher RPmean was seen in the patients' aortic arch and the DAo (Figure 5A(i)). Statistically significant differences between the two cohorts for RPmean were only observed in the LV ($1.080 \pm 0.427$ mmHg vs. $0.717 \pm 0.422$ mmHg, $p < 0.05$). Moreover, the controls demonstrated a higher PDmax in the AAo, aortic arch, DAo, and LV. There was no significant statistical difference observed between the two cohorts for PDmax. On the contrary, a higher RPmax for the controls was observed in the AAo and the LV, and it was also higher for the patients in the aortic arch and DAo (Figure 5A(ii)). Statistically significant differences between the two cohorts for RPmax were only found in the LV ($4.129 \pm 1.788$ mmHg vs. $2.432 \pm 1.491$ mmHg, $p < 0.05$). Lastly, a higher PDstd was observed in the AAo, DAo, and the LV of the controls and it was also slightly higher in the aortic arch of the patients. There was no significant statistical difference identified between the two cohorts for PDstd. In contrast, a higher RPstd was seen in the AAo and the LV of the controls, and it was also higher in the aortic arch and the DAo of the patients (Figure 5A(iii)). Again, statistically significant differences between the two cohorts for RPstd were only observed in the LV ($0.802 \pm 0.384$ mmHg vs. $0.435 \pm 0.335$ mmHg, $p < 0.05$).

For the right side of the heart, patients demonstrated a higher PDmean in all the vessels, including the LPA, RPA, MPA, and RV. Statistically significant differences between the two cohorts were observed for PDmean at RPA ($0.511 \pm 0.564$ mmHg vs. $2.690 \pm 2.788$ mmHg, $p < 0.05$), MPA ($1.176 \pm 0.572$ mmHg vs. $2.574 \pm 1.894$ mmHg, $p < 0.05$), and RV ($0.271 \pm 0.169$ mmHg vs. $0.641 \pm 0.675$ mmHg, $p < 0.05$). Similarly, patients demonstrated a higher RPmean in all four vessels (Figure 5B(i)). Statistically significant differences between the two cohorts for RPmean were only found in the MPA ($0.830 \pm 0.359$ mmHg vs. $2.289 \pm 1.399$ mmHg, $p < 0.05$). Additionally, PDmax was seen to be higher in patients in each vessel, including LPA, RPA, MPA, and the RV. Statistically significant differences between the two cohorts were witnessed for PDmax in the RPA ($1.506 \pm 1.851$ mmHg vs. $5.209 \pm 4.582$ mmHg, $p < 0.05$), MPA ($4.893 \pm 4.815$ mmHg vs. $9.778 \pm 10.098$ mmHg, $p < 0.05$), and RV ($2.483 \pm 2.176$ mmHg vs. $6.915 \pm 7.614$ mmHg, $p < 0.05$). Similarly, RPmax was higher in the patients' LPA, RPA, MPA, and RV (Figure 5B(ii)). Statistically significant differences between the two cohorts for RPmax were observed in the LPA ($2.222 \pm 1.294$ mmHg vs. $5.180 \pm 4.190$ mmHg, $p < 0.05$) and MPA ($2.073 \pm 0.742$ mmHg vs. $8.380 \pm 7.330$ mmHg, $p < 0.05$). Finally, patients demonstrated a higher PDstd in every vessel. Statistically significant differences between the two cohorts were found for PDstd at the RPA ($0.294 \pm 0.330$ mmHg vs. $1.100 \pm 0.991$ mmHg, $p < 0.05$), MPA ($0.949 \pm 0.57$ mmHg vs. $1.950 \pm 2.156$ mmHg, $p < 0.05$), and RV ($0.336 \pm 0.270$ mmHg vs. $0.896 \pm 0.817$ mmHg, $p < 0.05$). Likewise, a higher RPstd was observed in the patients' RPA, LPA, MPA, and RV (Figure 5B(iii)). Statistically significant differences between the two cohorts for RPstd were observed in the LPA ($0.429 \pm 0.258$ mmHg vs. $0.889 \pm 0.612$ mmHg, $p < 0.05$) and MPA ($0.447 \pm 0.196$ mmHg vs. $1.632 \pm 1.279$ mmHg, $p < 0.05$).

Figure 6 illustrates mean, max, and STD pressure measurements of the aorta, LV, PA, and RV reported as centerlines. For the left side of the heart, the controls demonstrated a higher PDmean in the AAo, aortic arch, DAo, and LV. Statistically significant differences between the two cohorts for PDmean was only identified in the AAo ($2.559 \pm 1.148$ mmHg vs. $1.682 \pm 1.164$ mmHg, $p < 0.05$). On the other hand, patients demonstrated a higher RPmean in the AAo, aortic arch, and DAo, whereas a slightly higher RPmean was seen in the LV of the controls (Figure 6A(i)). There was no significant statistical difference observed between the two cohorts for RPmax. Moreover, the controls demonstrated a higher PDmax

in the AAo, aortic arch, DAo, and LV. Then again, patients demonstrated a higher RPmax in the AAo, aortic arch, and DAo, whereas a slightly higher RPmax was seen in the LV of the controls (Figure 6A(ii)). No significant statistical difference was witnessed between the two cohorts for PDmax and RPmax. Lastly, a higher PDstd was observed in the AAo, aortic arch, DAo, and LV of the controls. Statistically significant differences between the two cohorts for PDstd were only found in the LV ($1.105 \pm 0.442$ mmHg vs. $1.037 \pm 0.603$ mmHg, $p < 0.05$). In contrast, a higher RPstd was seen in the patients' AAo, aortic arch, and DAo, and a slightly higher RPstd in the LV of the controls (Figure 6A(iii)). Statistically significant differences between the two cohorts for RPstd were only observed in the LV ($0.582 \pm 0.247$ mmHg vs. $0.518 \pm 0.379$ mmHg, $p < 0.05$).

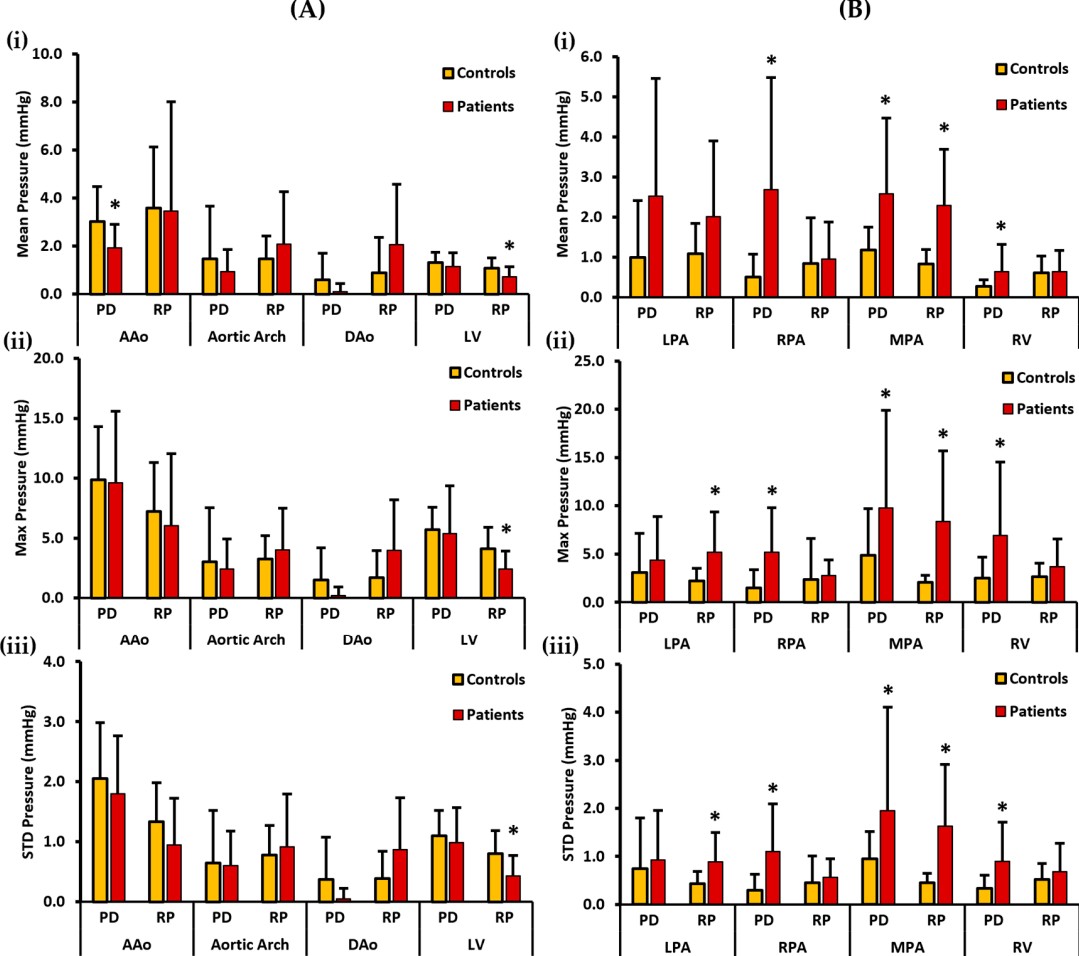

**Figure 5.** In vivo validation results: At peak systole, the pressure drop and relative pressures in the ascending aorta (AAo), aortic arch, descending aorta (DAo), left ventricle (LV), left pulmonary artery (LPA), right pulmonary artery (RPA), main pulmonary artery (MPA), and right ventricle (RV) were compared between the controls and the patients using vessel volumes as the reporting method. For the AAo, aortic arch, DAo, and LV (on the left), Panel (**A(i)**) shows the mean pressure drop (PD) and mean relative pressures (RP). Panel (**A(ii)**) shows the maximum PD and RP. Panel (**A(iii)**) shows the standard deviation of the PD and RP measurements. For the LPA, RPA, MPA, and RV (on the right), Panel (**B(i)**) shows the mean PD and RP. Panel (**B(ii)**) shows the max PD and RP, and Panel (**B(iii)**) shows the standard deviation of the PD and RP measurements. *: $p < 0.05$ between the controls and the patients. The controls showed a significantly higher PDmean in the AAo, and a significantly higher RPmean, RPmax, and RPstd in the LV compared to the patients. On the other hand, the patients showed a significantly higher PDmean, PDmax, and PDstd in the RPA, MPA, and the RV; a significantly higher RPmean, RPmax, and RPstd in the MPA; and a significantly higher RPmax and RPstd in the LPA compared to the controls.

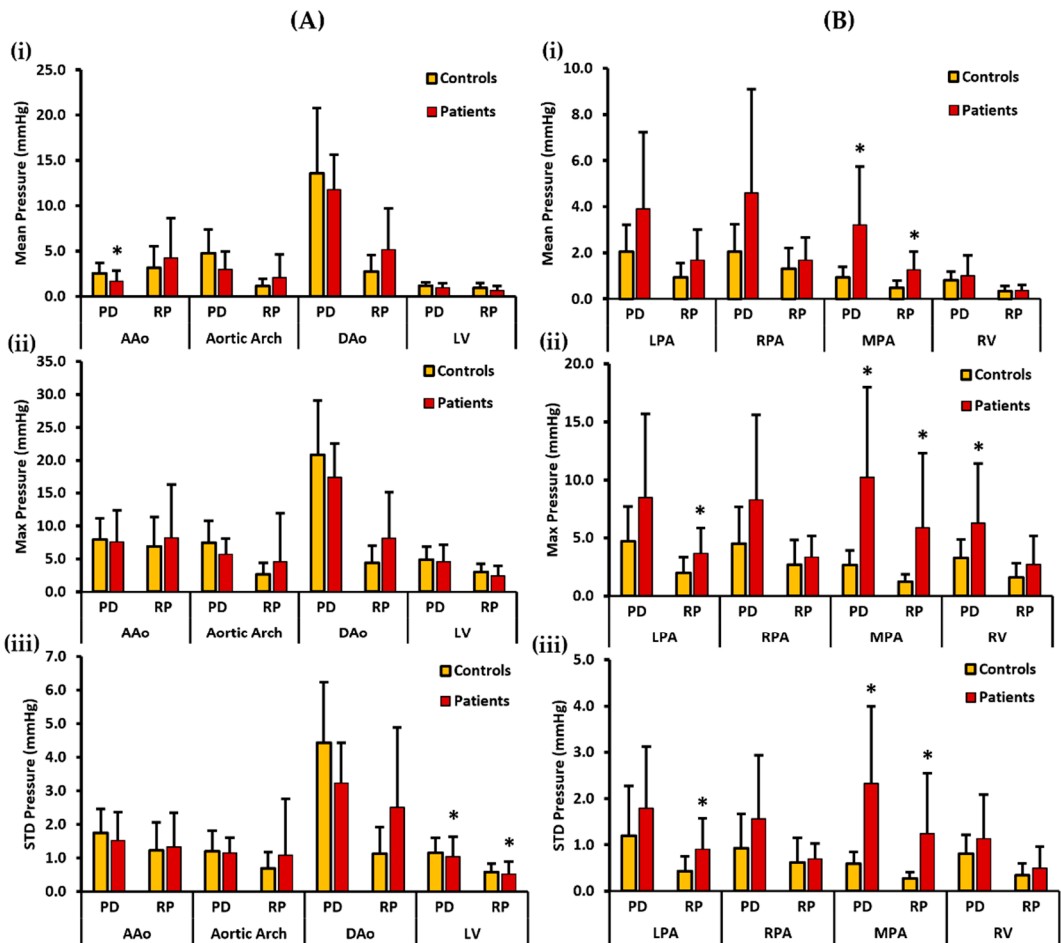

**Figure 6.** In vivo validation results: At peak systole, the pressure drop and relative pressures in the ascending aorta (AAo), aortic arch, descending aorta (DAo), left ventricle (LV), left pulmonary artery (LPA), right pulmonary artery (RPA), main pulmonary artery (MPA), and right ventricle (RV) were compared between the controls and the patients using the centerlines as the reporting method. For the AAo, aortic arch, DAo, and LV (on the left), Panel (**A(i)**) shows the mean pressure drop (PD) and mean relative pressures (RP). Panel (**A(ii)**) shows the maximum PD and RP. Panel (**A(iii)**) shows the standard deviation of the PD and RP measurements. For the LPA, RPA, MPA, and RV (on the right), Panel (**B(i)**) shows the mean PD and RP. Panel (**B(ii)**) shows the max PD and RP, and Panel (**B(iii)**) shows the standard deviation of the PD and RP measurements. *: $p < 0.05$ between the controls and the patients. The controls showed a significantly higher PDmean in the AAo, and a significantly higher PDstd and RPstd in the LV compared to the patients. Conversely, the patients show a significantly higher PDmean, PDmax, and PDstd in the MPA and a significantly higher PDmax in the RV than the controls. Furthermore, the patients showed a significantly higher RPmean, RPmax, and RPstd in the MPA and a significantly higher RPmax and RPstd in the LPA than the controls.

For the right side of the heart, the patients demonstrated a higher PDmean in all the vessels, including the LPA, RPA, MPA, and RV. Statistically significant differences between the two cohorts were observed for PDmean only in the MPA ($0.925 \pm 0.469$ mmHg vs. $3.211 \pm 2.534$ mmHg, $p < 0.05$). Similarly, patients demonstrated a higher RPmean in all vessels (Figure 6B(i)). Statistically significant differences between the two cohorts for RPmean were identified in the MPA ($0.477 \pm 0.318$ mmHg vs. $1.259 \pm 0.795$ mmHg, $p < 0.05$). PDmax was also seen to be higher in patients in each vessel, including LPA, RPA, MPA, and the RV. Statistically significant differences between the two cohorts were observed for PDmax in MPA ($2.656 \pm 1.281$ mmHg vs. $10.233 \pm 7.740$ mmHg, $p < 0.05$) and RV ($3.303 \pm 1.566$ mmHg vs. $6.274 \pm 5.140$ mmHg, $p < 0.05$). Similarly, RPmax was found

to be higher in each vessel of the patients (Figure 6B(ii)). Statistically significant differences between the two cohorts were witnessed for RPmax in the LPA (1.986 ± 1.340 mmHg vs. 3.682 ± 2.195 mmHg, $p < 0.05$) and MPA (1.256 ± 0.634 mmHg vs. 5.893 ± 6.384 mmHg, $p < 0.05$). Finally, patients demonstrated a higher PDstd in each vessel. Statistically significant differences between the two cohorts for PDstd were seen in the MPA (0.595 ± 0.251 mmHg vs. 2.330 ± 1.666 mmHg, $p < 0.05$). Likewise, patients demonstrated a higher RPstd in all vessels (Figure 6B(iii)). Statistically significant differences between the two cohorts for RPstd were observed in the LPA (0.435 ± 0.312 mmHg vs. 0.903 ± 0.676 mmHg, $p < 0.05$) and MPA (0.273 ± 0.136 mmHg vs. 1.239 ± 1.312 mmHg, $p < 0.05$).

Figure 7 depicts the mean, maximum, and standard deviation pressure measurements of the controls' aorta, LV, PA, and RV. For the left side of the heart, volumes demonstrated a higher PDmean in the AAo and the LV. On the other hand, centerlines demonstrated a higher PDmean in the aortic arch and the DAo. Statistically significant differences between the two cohorts for PDmean were observed in the aortic arch (1.465 ± 2.208 mmHg vs. 4.780 ± 2.635 mmHg, $p < 0.05$), DAo (0.597 ± 1.114 mmHg vs. 13.577 ± 7.211 mmHg, $p < 0.05$) and the LV (1.321 ± 0.418 mmHg vs. 0.992 ± 0.351 mmHg, $p < 0.05$). RPmean, on the contrary, was slightly higher in the AAo, aortic arch, and LV volumes. However, the centerline demonstrated a higher RPmean in the DAo of the controls. Therefore, statistically significant differences between the two cohorts for RPmean were found only in the DAo (0.885 ± 1.483 mmHg vs. 2.768 ± 1.793 mmHg, $p < 0.05$). Furthermore, volumes demonstrated a higher PDmax in the AAo and the LV. Centerlines, on the other hand, demonstrated a higher PDmax in the aortic arch and the DAo. Statistically significant differences between the two cohorts for PDmax were seen in the aortic arch (3.028 ± 4.496 mmHg vs. 7.456 ± 3.343 mmHg, $p < 0.05$), and the DAo (1.498 ± 2.701 mmHg vs. 20.812 ± 8.262 mmHg, $p < 0.05$).

On the contrary, a higher RPmax was observed in the volumes of the AAo, aortic arch, and LV, and in the centerline of the DAo. Statistically significant differences between the two cohorts for RPmax were identified in the DAo (1.715 ± 2.262 mmHg vs. 4.367 ± 2.609 mmHg, $p < 0.05$) and the LV (4.129 ± 1.788 mmHg vs. 2.992 ± 1.256 mmHg, $p < 0.05$). Finally, a higher PDstd was observed in the volume of the AAo. The centerlines demonstrated a higher PDstd in the aortic arch and DAo. PDstd was almost equal in the volume and centerline of the LV. Statistically significant differences between the two cohorts for PDstd was observed in the aortic arch (0.576 ± 0.874 mmHg vs. 1.163 ± 0.611 mmHg, $p < 0.05$), and the DAo (0.374 ± 0.703 mmHg vs. 4.429 ± 1.807 mmHg, $p < 0.05$). In contrast, volumes showed a higher RPstd in the AAo, aortic arch, and LV. RPstd was higher in the centerline of the DAo. Statistically significant differences between the two cohorts for RPstd were seen in the DAo (0.388 ± 0.455 mmHg vs. 1.128 ± 0.790 mmHg, $p < 0.05$) and the LV (0.802 ± 0.384 mmHg vs. 0.582 ± 0.247 mmHg, $p < 0.05$).

For the right side of the heart, centerlines demonstrated a higher PDmean in the LPA, RPA, and RV. A higher PDmean was found in the volume of the MPA. Statistically significant differences between the two cohorts were witnessed for PDmean in the LPA (0.994 ± 1.414 mmHg vs. 2.053 ± 1.166 mmHg, $p < 0.05$), RPA (0.511 ± 0.564 mmHg vs. 2.049 ± 1.191 mmHg, $p < 0.05$), and RV (0.271 ± 0.169 mmHg vs. 0.808 ± 0.374 mmHg, $p < 0.05$). Alternatively, volumes demonstrated a higher RPmean in the LPA, MPA, and RV. The centerline showed a higher RPmean in the RPA. Statistically significant differences between the two cohorts were perceived for RPmean in the RPA (0.847 ± 1.132 mmHg vs. 1.311 ± 0.888 mmHg, $p < 0.05$), MPA (0.830 ± 0.359 mmHg vs. 0.477 ± 0.318 mmHg, $p < 0.05$), and RV (0.608 ± 0.418 mmHg vs. 0.329 ± 0.224 mmHg, $p < 0.05$). Moreover, a higher PDmax was noted in the centerline of the LPA, RPA, and RV. A higher PDmax was observed in the volume of the MPA. Statistically significant differences between the two cohorts were observed for PDmax in the LPA (3.082 ± 4.042 mmHg vs. 4.702 ± 2.997 mmHg, $p < 0.05$), MPA (4.893 ± 4.815 mmHg vs. 2.656 ± 1.281 mmHg, $p < 0.05$), and RPA (1.506 ± 1.851 mmHg vs. 4.504 ± 3.160 mmHg, $p < 0.05$). Conversely, volumes demonstrated a higher RPmax in the LPA, MPA, and RV. The centerline demonstrated a higher

RPmax in the RPA. Statistically significant differences between the two cohorts were found for RPmax in the RPA ($2.376 \pm 4.235$ mmHg vs. $2.710 \pm 2.144$ mmHg, $p < 0.05$), MPA ($2.073 \pm 0.742$ mmHg vs. $1.256 \pm 0.634$ mmHg, $p < 0.05$), and RV ($2.674 \pm 1.391$ mmHg vs. $1.597 \pm 1.222$ mmHg, $p < 0.05$). Lastly, centerlines demonstrated a higher PDstd in the LPA, RPA, and RV. Volumes demonstrated a higher PDstd in the MPA. Statistically significant differences between the two cohorts were seen for PDstd in the LPA ($0.744 \pm 1.060$ mmHg vs. $1.196 \pm 1.072$ mmHg, $p < 0.05$), RPA ($0.294 \pm 0.330$ mmHg vs. $0.926 \pm 0.737$ mmHg, $p < 0.05$), and RV ($0.336 \pm 0.270$ mmHg vs. $0.808 \pm 0.410$ mmHg, $p < 0.05$). On the other hand, a higher RPstd was noted in the volumes of the MPA and RV, and in the centerline of the RPA. RPstd was almost equal in the volume and centerline of the LPA. Statistically significant differences between the two cohorts were identified for RPstd in the RPA ($0.450 \pm 0.556$ mmHg vs. $0.621 \pm 0.527$ mmHg, $p < 0.05$), MPA ($0.447 \pm 0.196$ mmHg vs. $0.273 \pm 0.136$ mmHg, $p < 0.05$), and RV ($0.524 \pm 0.333$ mmHg vs. $0.338 \pm 0.259$ mmHg, $p < 0.05$).

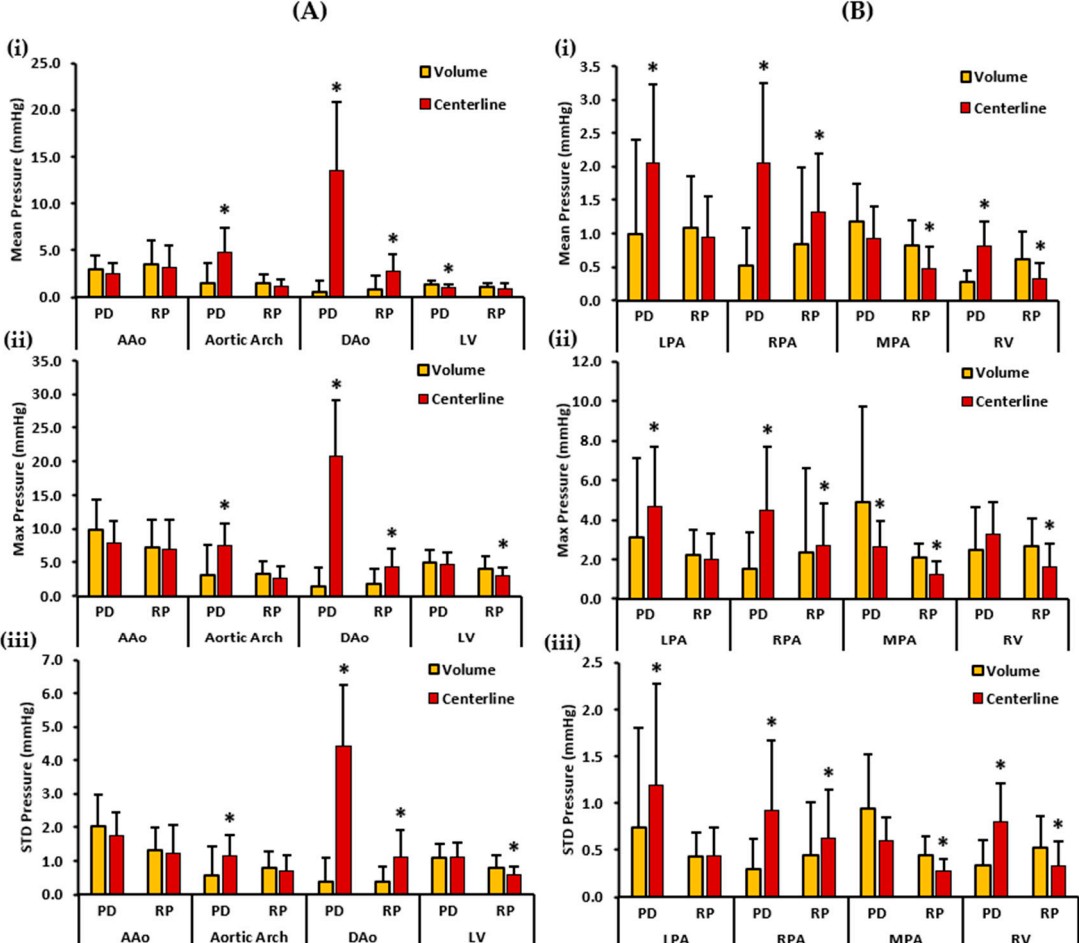

**Figure 7.** In vivo validation results: At peak systole, the pressure drop and relative pressures in the ascending aorta (AAo), aortic arch, descending aorta (DAo), left ventricle (LV), left pulmonary artery (LPA), right pulmonary artery (RPA), main pulmonary artery (MPA), and right ventricle (RV) of the controls were compared between the reporting methods (volumes and centerlines). For the AAo, aortic arch, DAo, and LV (on the left), Panel (**A**(**i**)) shows the mean pressure drop (PD) and mean relative pressures (RP). Panel (**A**(**ii**)) shows the maximum PD and RP. Panel (**A**(**iii**)) shows the standard deviation of the PD and RP measurements. For the LPA, RPA, MPA, and RV (on the right), Panel (**B**(**i**)) shows the mean PD and RP. Panel (**B**(**ii**)) shows the max PD and RP, and Panel (**B**(**iii**)) the shows standard deviation of the PD and RP measurements. *: $p < 0.05$ between the controls and

the patients. The volumes exhibited a significantly higher PDmean, RPmax, and RPstd in the LV compared to the centerlines. Furthermore, the volumes demonstrate a significantly higher PDmax, RPmean, RPmax, and RPstd in the MPA and a significantly higher RPmean, RPmax, and RPstd in the RV compared to the centerlines. On the contrary, the centerlines exhibited a significantly higher PDmean, PDmax, and PDstd in the aortic arch and DAo, and a significantly higher RPmean, RPmax, and RPstd in the DAo compared to the volumes. Additionally, the centerlines showed a higher PDmean, PDmax, and PDstd in the LPA and RPA, and a significantly higher PDmean and PDstd in the RV compared to the volumes. The centerlines also showed a higher RPmean, RPmax, and RPstd in the RPA compared to the volumes.

Figure 8 depicts the patients' mean, maximum, and standard deviation pressure measurements of the aorta, LV, PA, and RV. For the left side of the heart, the volumes demonstrated a higher PDmean in the AAo and the LV. On the other hand, the centerlines demonstrated a higher PDmean in the aortic arch and the DAo. Statistically significant differences between the two cohorts for PDmean were obtained in the aortic arch ($0.934 \pm 0.931$ mmHg vs. $2.984 \pm 1.992$ mmHg, $p < 0.05$) and DAo ($0.107 \pm 0.331$ mmHg vs. $11.800 \pm 3.832$ mmHg, $p < 0.05$). RPmean, on the contrary, was higher in the AAo, and DAo centerlines. However, RPmean was almost equal in the volume and centerline of the aortic arch and LV. Statistically significant differences between the two cohorts for RPmean were found only in the DAo ($2.057 \pm 2.523$ mmHg vs. $5.149 \pm 4.591$ mmHg, $p < 0.05$). Furthermore, volumes demonstrated a higher PDmax in the AAo and the LV. Centerlines, on the other hand, demonstrated a higher PDmax in the aortic arch and the DAo. Statistically significant differences between the two cohorts for PDmax were observed in the aortic arch ($2.417 \pm 2.531$ mmHg vs. $5.695 \pm 2.394$ mmHg, $p < 0.05$), and the DAo ($0.205 \pm 0.709$ mmHg vs. $17.370 \pm 5.167$ mmHg, $p < 0.05$). On the contrary, a higher RPmax was identified in the centerlines of the AAo, aortic arch, and DAo. RPmax was almost equal in the volume and centerline of the LV. Statistically significant differences between the two cohorts for RPmax were observed only in the DAo ($3.995 \pm 4.206$ mmHg vs. $8.100 \pm 7.057$ mmHg, $p < 0.05$). Finally, a higher PDstd was observed in the volume of the AAo. Centerlines demonstrated a higher PDstd in the aortic arch, DAo, and LV. Statistically significant differences between the two cohorts for PDstd were witnessed in the aortic arch ($0.602 \pm 0.577$ mmHg vs. $1.149 \pm 0.457$ mmHg, $p < 0.05$), and the DAo ($0.049 \pm 0.178$ mmHg vs. $3.230 \pm 1.194$ mmHg, $p < 0.05$). In contrast, centerlines showed a higher RPstd in all the vessels, including AAo, aortic arch, DAo, and LV. Statistically significant differences between the two cohorts for RPstd were observed only in the DAo ($0.869 \pm 0.867$ mmHg vs. $2.507 \pm 2.370$ mmHg, $p < 0.05$).

For the right side of the heart, the centerlines demonstrated a higher PDmean in all vessels, including LPA, RPA, MPA, and RV. Nevertheless, no statistically significant differences were found between the two cohorts for PDmean. Alternatively, volumes demonstrated a higher RPmean in the LPA, MPA, and RV. The centerline showed a higher RPmean in the RPA. Statistically significant differences between the two cohorts were observed for RPmean in the RPA ($0.887 \pm 0.931$ mmHg vs. $1.669 \pm 0.987$ mmHg, $p < 0.05$) and MPA ($2.289 \pm 1.399$ mmHg vs. $1.259 \pm 0.795$ mmHg, $p < 0.05$). Moreover, a higher PDmax was noted in the centerline of the LPA, RPA, MPA, and RV. Statistically significant differences between the two cohorts were found for PDmax only in the LPA ($4.357 \pm 4.509$ mmHg vs. $8.489 \pm 7.183$ mmHg, $p < 0.05$). Conversely, volumes demonstrated a higher RPmax in the LPA, MPA, and RV. The centerline demonstrated a higher RPmax in the RPA. Statistically significant differences between the two cohorts were observed for RPmax in the RPA ($1.676 \pm 1.621$ mmHg vs. $3.338 \pm 1.867$ mmHg, $p < 0.05$), and MPA ($8.380 \pm 7.330$ mmHg vs. $5.893 \pm 6.384$ mmHg, $p < 0.05$). Lastly, the centerlines demonstrated a higher PDstd in all vessels, including the LPA, RPA, MPA, and RV. Statistically significant differences between the two cohorts were identified for PDstd only in the LPA ($0.927 \pm 1.034$ mmHg vs. $1.794 \pm 1.335$ mmHg, $p < 0.05$). On the other hand, a higher RPstd was noted in the volumes of the MPA and RV; and in the centerline of the RPA. RPstd was almost equal in the

volume and centerline of the LPA. Statistically significant differences between the two cohorts were perceived for RPstd only in the RPA (0.411 ± 0.388 mmHg vs. 0.695 ± 0.338 mmHg, $p < 0.05$). Spearman's correlation was also calculated between standard clinical measurements and the two pressure mapping methods. However, no significant or strong correlation between the two was discovered.

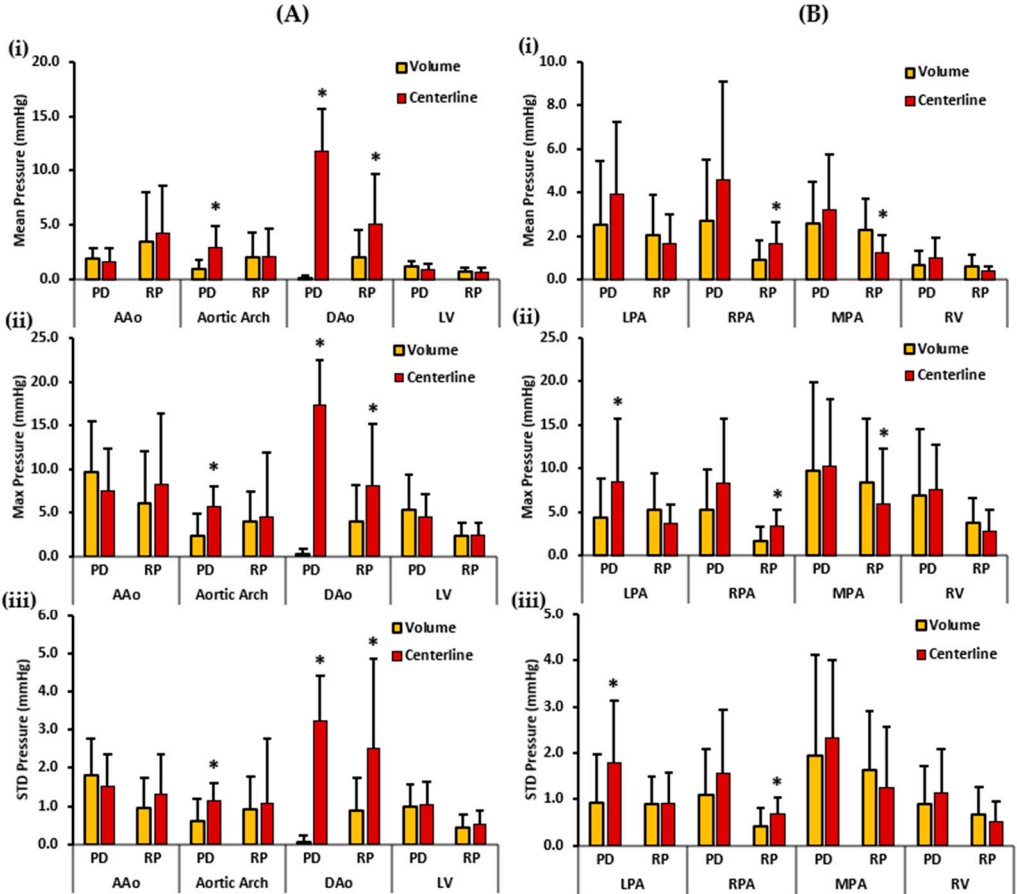

**Figure 8.** In vivo validation results: At peak systole, the pressure drop and relative pressures in the ascending aorta (AAo), aortic arch, descending aorta (DAo), left ventricle (LV), left pulmonary artery (LPA), right pulmonary artery (RPA), main pulmonary artery (MPA), and right ventricle (RV) of the patients were compared between the reporting methods (volumes and centerlines). For the AAo, aortic arch, DAo, and LV (on the left), Panel (**A**(**i**)) shows the mean pressure drop (PD) and mean relative pressures (RP). Panel (**A**(**ii**)) shows the maximum PD and RP. Panel (**A**(**iii**)) shows the standard deviation of the PD and RP measurements. For the LPA, RPA, MPA, and RV (on the right), Panel (**B**(**i**)) shows the mean PD and RP. Panel (**B**(**ii**)) shows the max PD and RP, and Panel (**B**(**iii**)) shows the standard deviation of the PD and RP measurements. *: $p < 0.05$ between controls and patients. Volumes exhibited significantly higher RPmean and RPmax in MPA compared to the centerlines. On the contrary, the centerlines indicate significantly higher PDmean, PDmax, and PDstd in the aortic arch and DAo; and significantly higher RPmean, RPmax, and RPstd in the DAo compared to volumes. Additionally, the centerlines showed a higher PDmax and PDstd in the LPA. The centerlines also showed a higher RPmean, RPmax, and RPstd in the RPA compared to the volumes.

## 4. Discussion

Our main findings showed: (1) the pressure drop and pressure gradient profiles followed velocity profiles whereas the relative pressure profiles aligned with the pressure recovery phenomenon; (2) a significantly higher pressure drop and relative pressure measurements in the MPA of the rTOF patients compared to the controls, reported as volumes

and centerlines; and (3) the volumes and centerlines are separate means of reporting pressures and do not reflect pressures in the same way.

Previous studies have demonstrated that 2D- and 4D-flow measurements in TOFs have a reasonable agreement [27–29]. In the context of this study, we evaluated flow only using 4D-flow MRI, and we derived the advanced hemodynamic parameter—pressure mapping. The term "pressure gradient" is often misused in the clinical literature, including clinical guidelines [30–32], where it is inaccurately used to describe the "pressure drop." Similarly, the term "relative pressure" is misused to refer to the "pressure gradient". Therefore, this study aimed to compare and solidify the understanding that pressure gradient, pressure drop, and relative pressures are all different pressure mapping methods that represent different information.

The simplified Bernoulli equation, which calculates the maximum pressure drop across a valve based on the maximum flow velocity at the vena contracta, has been found to overestimate the actual pressure drop in cases involving valve stenosis or downstream obstructions [33–35]. This discrepancy is primarily attributed to the phenomenon of pressure recovery, where a portion of the kinetic energy is converted back into pressure downstream of the stenosis. However, some kinetic energy is lost as thermal and acoustic energy. While this loss is minimal in laminar flow, it becomes significant under turbulent conditions. The simplified Bernoulli equation assumes that pressure recovery is negligible [36–40]. Our in vitro findings support this evidence. Pressure gradients estimated using the simplified Bernoulli equation overestimated the pressures compared to the pressure drop measurements; moreover, the profiles of the relative pressure measurements aligned with the pressure recovery phenomenon. However, as the number of VENCs increased, the RP profiles became unstable and failed to accurately depict the pressure recovery phenomenon. On the other hand, pressure gradient and pressure drop profiles remained stable and represented the maximum pressure drop at the obstruction.

The Reynolds numbers at the inlet and stenosis regions of the pipe indicated that the flow altered from being laminar to transitional to turbulent. The flow of a steady laminar stream in a circular pipe becomes unstable when the Reynolds number reaches 2000, and the transition into fully turbulent flow occurs at a Reynolds number of approximately 4000 [41]. Furthermore, as the SNR increased, the scatter plots revealed a progressive resemblance between the pressure measurements and the original pressure data. Additionally, the applied Gaussian filter demonstrated an efficient reduction of noise. Consistent results were observed across all five VENCs and for both pressure mapping methods. Moreover, with the increasing number of VENCs and turbulence, Bland–Altman analysis of the pressure drop method exhibited better agreement between the original and filtered pressure data compared to the relative pressure measurements. The pressure drops proved to be more robust under noise, as seen in Supplementary Materials.

Our in vivo study demonstrated a significantly higher PDmean, PDmax, PDstd, RPmean, RPmax, and RPstd in the MPA in the rTOF patients compared to the controls. The outcomes of this investigation were consistent with a study carried out by Sotelo et al. that showed pressure maps in rTOF patients had increased and that statistically significant pressure differences in the pulmonary system compared to healthy volunteers were present [42]. The elevated pressure measurements in the MPA could be attributed to the presence of pulmonary regurgitation (PR) in this patient group. PR and potentially residual or recurrent pulmonary stenosis are the most commonly seen complications in patients with rTOF [6]. Furthermore, our study also observed a higher PDmean, PDmax, PDstd, RPmean, RPmax, and RPstd in the LPA, RPA, and RV in the rTOF patients compared to the controls. The observed results could be attributed to anomalies that are predominantly located on the right side of the heart among individuals with rTOF. Following TOF surgery, several significant concerns may appear, including right ventricle enlargement and dysfunction, pulmonary regurgitation, as well as stenosis in both the right and left pulmonary arteries [43]. Patients with repaired TOFs have irregular flow patterns in their

right chambers; the majority of the inflow through the tricuspid valve is directed towards the RV's apex, mainly due to the PR jet [44].

Furthermore, this study observed higher values of PDmean, PDmax, and PDstd in the AAo, aortic arch, DAo, and LV in the controls compared to the patients. On the contrary, the controls showed higher values of RPmean, RPmax, and RPstd in the LV and in the volumes of AAo compared to the patients. The observed findings can be primarily attributed to the malformations predominantly occurring on the right side of the heart. Furthermore, the discrepancy may be attributed to lower velocities observed in these vessels among patients with rTOF compared to those in the control group. This indicates that individuals with rTOF may experience a disparity in velocities between the left and right cardiac chambers, resulting in heightened velocities on the right side and elevated pressure values in the RPA, LPA, MPA, and RV. Due to ventricular–ventricular interaction, the LV often develops a functional deterioration over time when there are impaired RV mechanics, as they share myofibers. Control left hearts are expected to function better than rTOF left hearts [45].

Unfortunately, studies investigating pressure mapping in rTOF patients are scarce. However, a substantial body of literature exists in the context of aortic valve disease and aortopathy [25,36,46]. Additionally, there is a limited literature on the assessment of 4D-flow-based pressure drop and relative pressures in rTOF patients. On the other hand, numerous studies demonstrated an elevated ventricular kinetic energy (KE), vorticity, wall shear stress (WSS), and turbulent KE in the patient cohort [47–49]. A study by Hirtler et al. reported a significantly higher intracardiac vorticity in patients with rTOF, as well as the association of a higher right vorticity with the regurgitant flow in the MPA [50]. WSS and energy loss are also known to be linked with pulmonary hemodynamic changes in the MPA and the RPA [51]. RV KE was also higher in the rTOF patients than in the healthy subjects [52], indicating that the RV must work harder in the rTOF patients to produce the same cardiac output as in the healthy subjects [53]. Vorticity, WSS, and KE were not evaluated or investigated in relation to the pressure drop or relative pressures in the current study. The association between pressure mapping and the aforementioned hemodynamic parameters is unclear and should be investigated. This study also compared the pressure drop and relative pressures to standard clinical measures, including LVEF, RVEF, LVEDVi, LVESVi, RVEDVi, and RVESVi. As no significant or strong correlation was observed between the pressure mapping methods and the standard clinical measures, this suggests that pressure mapping is an independent local measurement providing additional insights into the atypical flow patterns present in this group of patients.

Another novel finding of this study is that the volumes and centerlines are different methods and do not report pressures in the same way. Centerlines demonstrated a higher PDmean, PDmax, PDstd, RPmean, RPmax, and RPstd in the DAo and the RPA in both the controls and the patients. Moreover, the centerlines also demonstrated a higher PDmean, PDmax, and PDstd in the aortic arch, LPA, and RV of the controls and the patients; and were higher in the MPA of the patients. The study that proposed the 4D virtual catheter technique that mathematically mimics clinical invasive catheterization found that patients with bicuspid aortic valves had a higher viscous energy loss than healthy participants along the center volume of the aorta vessel [16]. A reason for the higher centerline measurements rather than volume measurements is the number of voxels. Centerline measurements are more local as they consider voxels only along the center of the vessel's volume. On the other hand, volume measurements are global; the reason for lower volume measurements could be owed to the effect of averaging over a greater number of voxels.

The repair of a TOF can produce multiple changes in the patient's hemodynamics, even in the case of a successful procedure [45,54]. The right ventricular function can be drastically impaired by pulmonary regurgitation by producing a chronic right-ventricular load increment, dilation of the right chambers, and a reduction in right heart performance. Pressure mapping based on 4D-flow MRI can overcome the limitations of a standard Doppler and characterize, more specifically, the anatomic locations with abnormal flow, pressure load, and vessel/chamber remodeling. These contributions may have implica-

tions for the indication of re-intervention during follow-up. Altered flow hemodynamics in rTOF are associated with a >3-fold increase in adverse cardiovascular events. For example, a peak right-ventricular outflow tract pressure gradient $\geq$ 25 mmHg has an HR = 3.69 [54]. Francois et al. [47] reported that rTOF patients showed an unbalanced flow distribution in the inferior and superior vena cava during the cardiac cycle, being greater during diastole. An increment of vortical flow in the right atria and right ventricle during diastole was also reported, along with the presence of altered flow in the pulmonary artery. All these observations were performed in consideration of their clinical relevance to the patients to evaluate the post-surgical alterations of geometry and hemodynamics. Furthermore, Hirtler et al. [50] demonstrated that the right ventricle and atrial vorticity were associated with chamber volumes and were also directly impacted by pulmonary regurgitation development.

There exist several possible limitations to our study. A significant limitation of this study is that only a small number of the rTOF subjects and controls were examined. More patients and healthy controls should be recruited to better understand how pressure mapping may have an impact on several vessels among patients in this cohort. An additional limitation of the study is the absence of data on inter- and intra-observer variability in calculating pressure maps. Because the NS equations calculate pressure gradients from both spatial and temporal velocity gradients, an insufficient spatial or temporal resolution will result in underestimating the pressure gradients and will lead to the loss of important fundamental or diagnostic pressure gradient features. Furthermore, a limited spatial resolution restricts small vessel analyses as high jet velocities and strong gradients at the jet boundary dominate stenotic blood flow. When the spatial resolution is sufficient, these strong gradients can be computed accurately. Recently introduced advanced imaging acceleration techniques, such as k-t under-sampling, compressed sensing, or radial under-sampling, are promising and have helped significantly reduce total scan times, allowing for a greater flexibility in spatial and temporal resolution selection [55]. Radial under-sampling overcomes the limitations of Cartesian 4D velocity mapping CMR by providing ample volume coverage with a high spatial resolution in reasonable scan times. Furthermore, radial acquisitions are also preferable to Cartesian acquisitions because they are less susceptible to motion artifacts [47]. Another limitation is that no validation against gold-standard invasive catheterization was performed in this study.

To extract the peak velocity, velocity maximum intensity projections (MIPs) were generated by masking the pre-processed 4D-flow MRI velocity field with the 3D segment. However, in our study, no noise filter was used to account for any false values caused by residual velocity aliasing or noise voxels. Rose et al. used a noise filter to exclude noise from the peak velocity assessment in their study, which used a vectorial vector containing voxel-wise velocity data [56].

Our in vitro validation results showed that pressure gradients, the pressure drop, and the relative pressure profiles are stable at lower VENCs. Conversely, with increasing turbulence and VENCs, the relative pressure profiles changed entirely and did not represent the pressure recovery phenomenon. Therefore, we recommend using the iterative approach over the multigrid-based PPE solver at higher VENCs. We did not evaluate the pressure maps according to VENC ranges in the patient population. However, this can be addressed by the multi-VENC acquisition strategies reported by Ha et al. [57]. Moreover, the reference points for the pressure drop calculation were manually selected in this study. A future alternative could be using a less operator-dependent automated reference detection approach using machine learning methods to reduce the variability in reference selection.

In our study, we included an experimental evaluation of pressure measurements. We acknowledged that particle image velocimetry is the experimental gold standard [58–60]. The conservation of mass principle is widely used in 4D-flow MRI as a quality control, and we used it in our data [13,61]. Theoretical values from cellar rotational flow [62], the Poiseuille and Lamb–Oseen equation [63], and the Hagen–Poiseuille equation [64] have been used for validation purposes of 4D-flow-derived metrics. In our study, we

considered only the experimental component as a validation and comparison of pressure methods. Furthermore, several studies proposed the integration of computational fluid dynamics (CFD) with 4D-flow MRI to overcome the spatial and temporal limitations of 4D-flow MRI [40,65,66]. In particular, finite element methods can be used to discretize the in vivo 4D-flow velocity field and estimate the pressure maps more accurately [10,67]. More recently, deep learning super-resolution approaches have been proposed to better capture the spatiotemporal characteristics of the 4D-flow MRI velocity field [68]. Our study did not include any of these novel approaches, but it should be considered in future assessments.

## 5. Conclusions

In conclusion, this study demonstrated that flow hemodynamics in rTOF can exhibit altered pressure maps. This study's results suggest that pressure mapping could be an independent biomarker for monitoring rTOF. Following the in vitro validation method, the pressure drops proved to be a more stable pressure mapping method than the method using relative pressures, as the flow loses its laminarity and becomes more turbulent. Further in vivo validation and longitudinal studies are needed to standardize a pressure mapping method that may provide further insight into rTOF patients' hemodynamics to improve patient care and clinical decisions.

**Supplementary Materials:** The following supporting information can be downloaded at: https://www.mdpi.com/article/10.3390/fluids8070196/s1, This file displays the Bland–Altman plot findings regarding the impact of Gaussian noise and filter on the pressure drop and relative pressure measurements. Figure S1: Measured pressure drop against distance, Y, for VENC 1.; Figure S2: Measured pressure drop against distance, Y, for VENC 2; Figure S3: Measured pressure drop against distance, Y, for VENC 3; Figure S4: Measured pressure drop against distance, Y, for VENC 4; Figure S5: Measured relative pressures against distance, Y, for VENC 1; Figure S6: Measured relative pressures against distance, Y, for VENC 2; Figure S7: Measured relative pressures against distance, Y, for VENC 3; Figure S8: Measured relative pressures against distance, Y, for VENC 4; Figure S9: Bland-Altman plots for pressure drop against distance, Y, for VENC 1; Figure S10: Bland-Altman plots for pressure drop against distance, Y, for VENC 2; Figure S11: Bland-Altman plots for pressure drop against distance, Y, for VENC 3; Figure S12: Bland-Altman plots for pressure drop against distance, Y, for VENC 4; Figure S13: Bland-Altman plots for relative pressures against distance, Y, for VENC 1; Figure S14: Bland-Altman plots for relative pressures against distance, Y, for VENC 2; Figure S15: Bland-Altman plots for relative pressures against distance, Y, for VENC 3; Figure S16: Bland-Altman plots for relative pressures against distance, Y, for VENC 4.

**Author Contributions:** Conceptualization, S.I.A. and J.G.; methodology, S.I.A. and J.G.; software, S.I.A. and J.G.; validation, S.I.A., D.P., K.A.M. and J.G.; formal analysis, S.I.A.; investigation, S.I.A. and J.G.; resources, J.G.; data curation, S.I.A.; writing—original draft preparation, S.I.A. and J.G.; writing—review and editing, D.P., K.A.M. and J.G.; visualization, S.I.A. and J.G.; supervision, J.G.; project administration, J.G.; funding acquisition, J.G. All authors have read and agreed to the published version of the manuscript.

**Funding:** This research was funded by The University of Calgary, URGC SEM #1054341; J.G. start-up funding. Unrestricted research funding was provided by Siemens Healthineers. We acknowledge the support of the Natural Science and Engineering Research Council of Canada/Conseil de recherche en science naturelles et en génie du Canada, RGPIN-2020-04549 and DGECR-2020-00204.

**Institutional Review Board Statement:** The study was conducted according to the guidelines of the Declaration of Helsinki and approved by the Conjoint Health Research Ethics Board of the University of Calgary (REB13-0902 was approved on 18 June 2014 and is currently active).

**Informed Consent Statement:** Written informed consent was obtained from all subjects involved in the study.

**Data Availability Statement:** The anonymized data presented in this study are available upon request from the corresponding author. The data are not publicly available due to privacy and ethical restrictions.

**Conflicts of Interest:** The authors declare no conflict of interest.

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
