# Peer review of "Repaired Tetralogy of Fallot Pressure Assessment: Insights from 4D-Flow Pressure Mapping"

_fluids, doi:10.3390/fluids8070196_

Round 1
Reviewer 1 Report
Dear Authors. Thank you for the opportunity to comment on: Repaired Tetralogy of Fallot Hemodynamics: Insights from 4D-2 Flow Pressure Mapping. First of all this is a very interesting paper and relatively small group of patients does not diminish the value of the manuscript. However the manuscript need to undergo minor revisions: The part “In-vitro validation using a stenosis phantom was 114 performed, followed by an in-vivo validation between (i) healthy subjects and patients 115 with rTOF reported using volumes and centerlines and (ii) volume and centerline data of 116 controls and patients.” Should be in the beginning of methods section. Conclusion shoud be shorter, “Pressure drops and relative pressures in the LPA, RPA, MPA, and 682 RV were higher in rTOF patients compared to the controls.” – there is the result not conclucion.
Author Response
Ms. Ref. No: fluids-2400871
Manuscript title: Repaired Tetralogy of Fallot Pressure Assessment: Insights from 4D-Flow Pressure Mapping
We thank reviewers for their valuable comments and suggestions regarding our study and manuscript. Point-by-point response follows, and changes are tracked in the annotated version of the manuscript. Please remark the article title changed.
Response to Reviewer 1
R1-C#1: The part “In-vitro validation using a stenosis phantom was performed, followed by an in-vivo validation between (i) healthy subjects and patients with rTOF reported using volumes and centerlines and (ii) volume and centerline data of controls and patients.” Should be in the beginning of methods section.”
We appreciate this suggestion. We fixed this section and moved this paragraph to the beginning of the methods section.
R1-C#2: Conclusion should be shorter, “Pressure drops and relative pressures in the LPA, RPA, MPA, and RV were higher in rTOF patients compared to the controls.” – there is the result not conclusion.
We agree with the reviewer. This sentence was removed from the conclusion.
Reviewer 2 Report
The authors studied the research study on Tetralogy of Fallot (TOF), a common congenital heart defect. The study aimed to compare pressure mapping and relative pressures in patients with repaired TOF (rTOF) and healthy individuals. The researchers used 4D-flow magnetic resonance imaging (MRI) to obtain pressure mapping data. In-vitro and in-vivo validations were performed to assess the stability of pressure drop and relative pressures. The results showed that pressure drops and relative pressures in the main pulmonary artery (MPA) were higher in rTOF patients compared to controls. The in-vitro validation confirmed that pressure drops were more stable as flow became turbulent. The study concludes that pressure mapping can provide valuable insights into the altered hemodynamics in rTOF patients, which can aid in improving patient care and clinical decisions.
The article needs a minor revision:
Comments:
1): The authors used the word Hemodynamics in the title, but in the abstract or introduction I didn't find any fruitful discussion.
2): Provide a comparison paragraph in the introduction about Hemodynamics, Hydrodynamics, and connection between them, the authors could use the literature (i): OpenFOAM for computational hydrodynamics using finite volume method, Micropolar Dusty Fluid: Coriolis Force Effects on Dynamics of MHD Rotating Fluid When Lorentz Force Is Significant.
Morphological nanolayer impact on hybrid nanofluids flow due to dispersion of polymer/CNT matrix nanocomposite material
3): How can pressure mapping contribute to patient care and clinical decisions in rTOF?
4): What are the potential implications of altered flow hemodynamics in rTOF patients?
5): The authors need to improve the discussion section by providing more physical discussion to the graphical results, the authors are suggested to follow the physical discussion or include them in the discussion section, i.e., The shortfall and rise in energy deposition and combustion via OpenFOAM.
6): Reduce the plagiarism.
Author Response
Ms. Ref. No: fluids-2400871
Manuscript title: Repaired Tetralogy of Fallot Pressure Assessment: Insights from 4D-Flow Pressure Mapping
We thank reviewers for their valuable comments and suggestions regarding our study and manuscript. Point-by-point response follows, and changes are tracked in the annotated version of the manuscript. Please remark the article title changed.
Response to Reviewer 2
R2-C#1: “The authors used the word Hemodynamics in the title, but in the abstract or introduction I didn't find any fruitful discussion.”
We appreciate this observation. To better express the content of the study we have change the word hemodynamics to pressure assessment in the title.
R2-C#2: “Provide a comparison paragraph in the introduction about Hemodynamics, Hydrodynamics, and connection between them, the authors could use the literature (i): OpenFOAM for computational hydrodynamics using finite volume method, Micropolar Dusty Fluid: Coriolis Force Effects on Dynamics of MHD Rotating Fluid When Lorentz Force Is Significant. Morphological nanolayer impact on hybrid nanofluids flow due to dispersion of polymer/CNT matrix nanocomposite material.”
We appreciate this suggestion. We had difficulty understanding the context of the proposed literature which seemed off topic for this study. However, we considered the CFD aspect as it is a relevant addition to the manuscript discussion and limitations. Particularly considering recent studies integrating finite-element analysis with 4D-flow MRI, comparisons with CFD for validation purposes, and novel approaches using DL super-resolution. We have added following paragraph in the discussion, limitations section:
“In our study, we included an experimental evaluation for pressure measurements. We acknowledged that particle image velocimetry is considered the experimental gold-standard (1-3). The conservation of mass’ principle is widely used in 4D-flow MRI as a quality control and we used it in our data (4, 5). Theoretical values from cellar rotational flow (6), Poiseuille and Lamb-Oseen equation (7), and Hagen-Poiseuille equation (8) have been used for validation purposes of 4D-flow derived metrics. In our study, we considered only the experimental component as validation and comparison of pressure methods. Furthermore, several studies proposed the integration of computational fluid dynamics (CFD) with 4D-flow MRI to overcome spatial and temporal limitations of 4D-flow MRI (9-11). In particular, finite element methods can be used to discretize the in-vivo 4D-flow velocity field and used to estimate pressure maps more accurately (12, 13). More recently, deep learning super-resolution approaches have been proposed to capture better the spatiotemporal characteristics of the 4D-flow MRI velocity field (14). Our study did not include any of these novel approaches but it must be considered for future assessments.”
- Brindise MC, Rothenberger S, Dickerhoff B, et al. Multi-modality cerebral aneurysm haemodynamic analysis: in vivo 4D flow MRI, in vitro volumetric particle velocimetry and in silico computational fluid dynamics. J R Soc Interface 2019; 16:20190465. 10.1098/rsif.2019.0465
- Medero R, Ruedinger K, Rutkowski D, Johnson K, Roldán-Alzate A. In vitro assessment of flow variability in an intracranial aneurysm model using 4d flow mri and tomographic piv. Ann Biomed Eng 2020; 48:2484–2493. 10.1007/s10439-020-0
- Kitajima HD, Sundareswaran KS, Teisseyre TZ, Astary GW, Parks WJ, Skrinjar O, Oshinski JN, Yoganathan AP. Comparison of particle image velocimetry and phase contrast MRI in a patient-specific extracardiac total cavopulmonary connection. J Biomech Eng. 2008 Aug;130(4):041004. doi: 10.1115/1.2900725. PMID: 18601446.
- Dyverfeldt, P., Bissell, M., Barker, A.J. et al. 4D flow cardiovascular magnetic resonance consensus statement. J Cardiovasc Magn Reson 17, 72 (2015). https://doi.org/10.1186/s12968-015-0174-5
- Zhong L, Schrauben EM, Garcia J, Uribe S, Grieve SM, Elbaz MSM, Barker AJ, Geiger J, Nordmeyer S, Marsden A, Carlsson M, Tan RS, Garg P, Westenberg JJM, Markl M, Ebbers T. Intracardiac 4D Flow MRI in Congenital Heart Disease: Recommendations on Behalf of the ISMRM Flow & Motion Study Group. J Magn Reson Imaging. 2019 Sep;50(3):677-681. doi: 10.1002/jmri.26858. Epub 2019 Jul 17. PMID: 31317587.
- Garcia, J., Larose, E., Pibarot, P., and Kadem, L. (October 24, 2013). "On the Evaluation of Vorticity Using Cardiovascular Magnetic Resonance Velocity Measurements." ASME. J Biomech Eng. December 2013; 135(12): 124501. https://doi.org/10.1115/1.4025385
- Sotelo J, Urbina J, Valverde I, Mura J, Tejos C, Irarrazaval P, Andia ME, Hurtado DE, Uribe S. Three-dimensional quantification of vorticity and helicity from 3D cine PC-MRI using finite-element interpolations. Magn Reson Med. 2018 Jan;79(1):541-553. doi: 10.1002/mrm.26687. Epub 2017 Mar 31. PMID: 28370386.
- Fukuyama A, Isoda H, Morita K, Mori M, Watanabe T, Ishiguro K, Komori Y, Kosugi T. Influence of Spatial Resolution in Three-dimensional Cine Phase Contrast Magnetic Resonance Imaging on the Accuracy of Hemodynamic Analysis. Magn Reson Med Sci. 2017 Oct 10;16(4):311-316. doi: 10.2463/mrms.mp.2016-0060. Epub 2017 Jan 30. PMID: 28132996; PMCID: PMC5743522.
- Casas B, Lantz J, Dyverfeldt P, Ebbers T. 4D Flow MRI-based pressure loss estimation in stenotic flows: Evaluation using numerical simulations. Magn Reson Med. 2016 Apr;75(4):1808-21. doi: 10.1002/mrm.25772. Epub 2015 May 28. PMID: 26016805.
- Sadeghi, R., Tomka, B., Khodaei, S. et al. Impact of extra-anatomical bypass on coarctation fluid dynamics using patient-specific lumped parameter and Lattice Boltzmann modeling. Sci Rep 12, 9718 (2022). https://doi.org/10.1038/s41598-022-12894-y
- Canstein C, Cachot P, Faust A, Stalder AF, Bock J, Frydrychowicz A, Küffer J, Hennig J, Markl M. 3D MR flow analysis in realistic rapid-prototyping model systems of the thoracic aorta: comparison with in vivo data and computational fluid dynamics in identical vessel geometries. Magn Reson Med. 2008 Mar;59(3):535-46. doi: 10.1002/mrm.21331. PMID: 18306406.
- Nolte D, Urbina J, Sotelo J, Sok L, Montalba C, Valverde I, Osses A, Uribe S, Bertoglio C. Validation of 4D Flow based relative pressure maps in aortic flows. Med Image Anal. 2021 Dec;74:102195. doi: 10.1016/j.media.2021.102195. Epub 2021 Aug 15. PMID: 34419837.
- Lamata P, Pitcher A, Krittian S, Nordsletten D, Bissell MM, Cassar T, Barker AJ, Markl M, Neubauer S, Smith NP. Aortic relative pressure components derived from four-dimensional flow cardiovascular magnetic resonance. Magn Reson Med. 2014 Oct;72(4):1162-9. doi: 10.1002/mrm.25015. Epub 2013 Nov 18. PMID: 24243444; PMCID: PMC4024466.
- Fathi MF, Perez-Raya I, Baghaie A, Berg P, Janiga G, Arzani A, D'Souza RM. Super-resolution and denoising of 4D-Flow MRI using physics-Informed deep neural nets. Comput Methods Programs Biomed. 2020 Dec;197:105729. doi: 10.1016/j.cmpb.2020.105729. Epub 2020 Sep 15. PMID: 33007592.
R2-C#3: “How can pressure mapping contribute to patient care and clinical decisions in rTOF?”
The repair of TOF can produce multiple changes in the patient’s hemodynamics even in the case of a successful procedure (15, 16). The right ventricular function can be drastically impaired by pulmonary regurgitation by producing a chronic right ventricular load increment, dilation of the right chambers, and a reduction of the right heart performance. Pressure mapping based on 4D-flow MRI can overcome the limitations of standard Doppler and characterize more specifically anatomic locations with the abnormal flow, pressure load, and vessel/chamber remodeling. These contributions may have implications for the indication of re-intervention during follow-up.
We have included this comment in the discussion section.
- Geva, T. Repaired tetralogy of Fallot: the roles of cardiovascular magnetic resonance in evaluating pathophysiology and for pulmonary valve replacement decision support. J Cardiovasc Magn Reson 13, 9 (2011). https://doi.org/10.1186/1532-429X-13-9
- Latus H, Stammermann J, Voges I, Waschulzik B, Gutberlet M, Diller GP, Schranz D, Ewert P, Beerbaum P, Kühne T, Sarikouch S; German Competence Network for Congenital Heart Defects Investigators *. Impact of Right Ventricular Pressure Load After Repair of Tetralogy of Fallot. J Am Heart Assoc. 2022 Apr 5;11(7):e022694. doi: 10.1161/JAHA.121.022694. Epub 2022 Mar 18. PMID: 35301850; PMCID: PMC9075442.
R2-C#4: “What are the potential implications of altered flow hemodynamics in rTOF patients?”
Altered flow hemodynamics in rTOF is associated with a >3-fold increase of adverse cardiovascular events. For example, a peak right ventricular outflow tract pressure gradient ≥ 25 mmHg has an HR = 3.69 (16). Francois et al. (17) reported that rTOF patients showed and unbalanced flow distribution in the inferior and superior vena cava during the cardiac cycle, being greater during diastole. An increment of vortical flow in the right atria and right ventricle during diastole was also reported along with the presence of altered flow in the pulmonary artery. All these observations were performed in the consideration of the clinical relevance for the patients to evaluate the post-surgical alterations of geometry and hemodynamics. Furthermore, Hirtler et al. (18) demonstrated that right ventricle and atrial vorticity were associated with chamber volumes and were directly impacted by the development of pulmonary regurgitation.
We have integrated these aspects to the discussion.
- François CJ, Srinivasan S, Schiebler ML, Reeder SB, Niespodzany E, Landgraf BR, Wieben O, Frydrychowicz A. 4D cardiovascular magnetic resonance velocity mapping of alterations of right heart flow patterns and main pulmonary artery hemodynamics in tetralogy of Fallot. J Cardiovasc Magn Reson. 2012 Feb 7;14(1):16. doi: 10.1186/1532-429X-14-16. PMID: 22313680; PMCID: PMC3305663.
- Hirtler D, Garcia J, Barker AJ, Geiger J. Assessment of intracardiac flow and vorticity in the right heart of patients after repair of tetralogy of Fallot by flow-sensitive 4D MRI. Eur Radiol. 2016 Oct;26(10):3598-607. doi: 10.1007/s00330-015-4186-1. Epub 2016 Jan 8. PMID: 26747260; PMCID: PMC4938791.
R2-C#5: “The authors need to improve the discussion section by providing more physical discussion to the graphical results, the authors are suggested to follow the physical discussion or include them in the discussion section, i.e., The shortfall and rise in energy deposition and combustion via OpenFOAM.”
Please see R2-C#2
R2-C#6: “Reduce the plagiarism. I will do a plagiarism check once after all the changes have been made.”
We have verified the entire manuscript for plagiarism. We used the MDPI similarity report to reduce it and dedicated on-line tools (Grammarly, dupli checker, and iThenticate).
Reviewer 3 Report
This is an interesting study. I recommend acceptance after addressing a few major comments as appended below:
line 54: an excess space is found.
line 108-116: references are required
line 170-171: A total of 36 subjects, including 18 rTOF patients (age: 32 ± 10, 6 females) and 18 170 controls (age: 37 ± 14, 7 females), were recruited retrospectively. The reviewer's comment: Please indicate whether these values are mean with the standard deviation or SEM.
line 187-253: Please support the methodology part with the appropriate references.
Another recommendation: a schematic diagram may be needed to enhance the presentation of this manuscript.
line 284: in the result section, the authors should highlight the p-value in the text.
line 373, 439, 478, 488: Each figure should have a clear legend referring to the significant findings. Please address this point in the whole manuscript.
line 544-558: This paragraph should be edited. Please first highlight your main findings. Next, try to discuss these findings.
Minor editing of English language required
Author Response
Ms. Ref. No: fluids-2400871
Manuscript title: Repaired Tetralogy of Fallot Pressure Assessment: Insights from 4D-Flow Pressure Mapping
We thank reviewers for their valuable comments and suggestions regarding our study and manuscript. Point-by-point response follows, and changes are tracked in the annotated version of the manuscript. Please remark the article title changed.
Response to Reviewer 3
R3-C#1: “line 54: an excess space is found.”
Fixed.
R3-C#2: “line 108-116: references are required”
We added corresponding references.
.
R3-C#3: “line 170-171: A total of 36 subjects, including 18 rTOF patients (age: 32 ± 10, 6 females) and 18 170 controls (age: 37 ± 14, 7 females), were recruited retrospectively. The reviewer's comment: Please indicate whether these values are mean with the standard deviation or SEM.”
Fixed.
R3-C#4: “line 187-253: Please support the methodology part with the appropriate references.”
We have added the corresponding references to support the methodology.
R3-C#5: “Another recommendation: a schematic diagram may be needed to enhance the presentation of this manuscript”.
We added a new diagram to better describe the methodology used in the study.
R3-C#6: “Aline 284: in the result section, the authors should highlight the p-value in the text.”
For the in-vitro section, Bland-Altman Analysis was performed. No statistical analysis leading to a p-value was performed. P-value was properly presented when it was needed.
R3-C#7: “Aline 373, 439, 478, 488: Each figure should have a clear legend referring to the significant findings. Please address this point in the whole manuscript.”
Fixed. Added significant findings to the legends.
R3-C#8: “line 544-558: This paragraph should be edited. Please first highlight your main findings. Next, try to discuss these findings.”
Fixed. Edited the paragraph.
Round 2
Reviewer 3 Report
The authors addressed all comments. I think it could be acceptable.
A minor English editing is needed.
Author Response
We performed a more detail language revision and further analysis to reduce similarity score in an effort to improve the manuscript.